# Endothelial discoidin domain receptor 1 senses flow to modulate YAP activation

Jiayu Liu[1,2,6], Chuanrong Zhao[1,2,3,6], Xue Xiao[4], Aohan Li[5], Yueqi Liu[1,2], Jianan Zhao[1,2], Linwei Fan[1,2], Zhenhui Liang[1,2], Wei Pang[1], Weijuan Yao[1], Wei Li[4] ✉ & Jing Zhou[1,2] ✉

Mechanotransduction in endothelial cells is critical to maintain vascular homeostasis and can contribute to disease development, yet the molecules responsible for sensing flow remain largely unknown. Here, we demonstrate that the discoidin domain receptor 1 (DDR1) tyrosine kinase is a direct mechanosensor and is essential for connecting the force imposed by shear to the endothelial responses. We identify the flow-induced activation of endothelial DDR1 to be atherogenic. Shear force likely causes conformational changes of DDR1 ectodomain by unfolding its DS-like domain to expose the buried cysteine-287, whose exposure facilitates force-induced receptor oligomerization and phase separation. Upon shearing, DDR1 forms liquid-like biomolecular condensates and co-condenses with YWHAE, leading to nuclear translocation of YAP. Our findings establish a previously uncharacterized role of DDR1 in directly sensing flow, propose a conceptual framework for understanding upstream regulation of the YAP signaling, and offer a mechanism by which endothelial activation of DDR1 promotes atherosclerosis.

Vascular endothelial cells (ECs) lining the inner surface of blood vessels constantly bears the frictional forces (wall shear stress) derived from the hemodynamic flow. Shear stress can be either detrimental or beneficial, depending on the flow patterns determined by vessel geometry[1,2]. Disturbed flow with a low and reciprocating shear stress (oscillatory shear, OS) in arterial branches and curvatures upregulates pro-inflammatory molecules in ECs and damages the endothelial function, ultimately leading to atherosclerosis. In contrast, laminar flow with a high and unidirectional shear stress (pulsatile shear or laminar shear, PS or LS) in the straight parts of the arteries induces the alignment of ECs to the direction of flow and elicits an anti-inflammatory and atheroprotective effect on the vasculature[3]. As the primary initiators of mechanotransduction cascades that drive the

development of either atheroprone or atheroprotective endothelial phenotypes, flow mechanosensors in ECs have become the focus of an increasing number of studies aimed at identifying them[4–11].

The discoidin domain receptor 1 (DDR1) tyrosine kinase is fundamental for proper embryonic development and organogenesis and also implicated in the progression of several diseases including various cancers, atherosclerosis and fibrotic diseases[12]. DDR1 is mechanoresponsive to extracellular matrix (ECM) stiffness[13–16], but it remains unexplored whether and how DDR1 mediates flow sensation in ECs. Structurally, DDR1 is composed of N-terminal extracellular globular discoidin (DS) domain and DS-like domain, followed by an extracellular juxtamembrane (JM) region, a transmembrane helix, a large cytosolic JM region, and a C-terminal tyrosine kinase domain[17]. DDR1

[1]Department of Physiology and Pathophysiology, School of Basic Medical Sciences, State Laboratory of Vascular Homeostasis and Remodeling, Peking University, Beijing 100191, China. [2]National Health Commission Key Laboratory of Cardiovascular Molecular Biology and Regulatory Peptides, Beijing Key Laboratory of Cardiovascular Receptors Research, Peking University, Beijing 100191, China. [3]Key Laboratory of Biorheological Science and Technology, Ministry of Education, College of Bioengineering, Chongqing University, Chongqing 400044, China. [4]National Laboratory of Biomacromolecules and Key Laboratory of Epigenetic Regulation and Intervention, Institute of Biophysics, Chinese Academy of Sciences, Beijing 100101, China. [5]National Laboratory for Condensed Matter Physics and Key Laboratory of Soft Matter Physics, Institute of Physics, Chinese Academy of Sciences, Beijing 100190, China. [6]These authors contributed equally: Jiayu Liu, Chuanrong Zhao. ✉e-mail: weili007@iphy.ac.cn; jzhou@bjmu.edu.cn

has been found to exist as stable and noncovalently linked dimers that form during biosynthesis. Ligand-independent covalent dimerization via a disulfide bond has been documented to be mediated by the transmembrane region[17]. Collagen binding to the DS domain of dimerized DDR1 results in receptor activation through oligomerization and the subsequent autophosphorylation of cytoplasmic tyrosine residues[18]. The roles of the DS-like domain played in DDR1 activation are poorly defined.

Disturbed flow increases whereas laminar flow inhibits the endothelial Yes-associated protein (YAP) activity in terms of dephosphorylation and translocation into the nucleus[19,20]. Activation of YAP signaling promotes endothelial inflammation and vascular dysfunction, resulting in atherogenesis[19,20]. A major unanswered question in YAP signaling concerns how the external mechanical cues are transduced and signaled the upstream regulator of YAP. Studies to date have implicated several mechanosensory molecules and complexes including integrins and focal adhesion proteins[21,22]. Recently, we and others reported that DDR1 mediates the ECM stiffness-induced YAP activation in vascular smooth muscle cells[15,16]. However, whether endothelial DDR1 senses flow to coordinate YAP is unclear.

In this study, we show that DDR1 acts as a direct mechanosensor in ECs, regulating the cellular responses to shear flow and hence the site-specific distribution of atherosclerosis. We further identify the role and the mechanism of its DS-like domain in force-induced oligomerization of DDR1 and reveal that force-activated DDR1 forms liquid-like biomolecular condensates with 14-3-3ε protein (YWHAE) for inhibiting phosphorylation and cytoplasmic retention of YAP.

## Results

### DDR1 mediates the endothelial responses to fluid shear stress

To explore whether DDR1 is required for the EC responses to shear flow, human umbilical vein ECs (HUVECs) transfected with siRNA specifically recognizing *DDR1* or scrambled siRNA (Supplementary Fig. 1A, B) were seeded on gelatin-coated slides and subjected to PS (12 ± 4 dyn/cm2, 1 Hz), LS (12 dyn/cm2), or OS (0.5 ± 4 dyn/cm2, 1 Hz) by using a parallel-plate flow apparatus (Fig. 1A). Immunofluorescence staining of F-actin and analysis of its orientation indicated that the cell alignment with the direction of shear flow was compromised in DDR1-knockdown cells (Fig. 1B and Supplementary Fig. 1C). To determine whether DDR1 is sufficient to confer the mechanosensitivity in heterologous cells, NIH3T3 fibroblast cells that barely express DDR1[23] were infected with DDR1-EGFP recombinant adenovirus or control virus and subjected to shear exposure. Surprisingly, DDR1-expressing NIH3T3 cells aligned along the direction of PS or LS (but not OS), whereas no such phenomenon was observed in the control cells (Fig. 1C and Supplementary Fig. 1D). Analysis of the mRNA levels of anti-inflammatory transcription factors (KLF2 and KLF4) and proinflammatory molecules (MCP1 and E-selectin) in sheared HUVECs revealed that knockdown of DDR1 attenuated the atheroprone OS-induced upregulation of MCP1 and E-selectin and reduced the expression of KLF2 and KLF4 in the PS condition (Fig. 1D). To investigate the biological significance of our findings, we generated the inducible EC-specific *Ddr1* knockout mice *Ddr1*iECKO (*Ddr1*flox/flox, *Cdh5-CreER*T2+) (Supplementary Fig. 2A, B) and employed the mice to test the role of endothelial DDR1 in disturbed flow-accelerated vascular inflammation and atherosclerosis. *Ddr1*WT (*Ddr1*flox/flox, *Cdh5-CreER*T2-) and *Ddr1*iECKO mice were injected once with adeno-associated-virus-8 (AAV8)-overexpressing PCSK9 (AAV8-PCSK9) via the tail vein and fed on a Western diet to induce hyperlipidemia, a potent risk factor for atherosclerosis. A partial ligation surgery[24] of the left carotid artery was performed to introduce disturbed blood flow into the left common carotid artery (Fig. 1E). At 4-week post-operation, the *Ddr1*iECKO mice exhibited significantly decreased atherosclerotic plaque compared with the *Ddr1*WT mice (Fig. 1F–H), although the body weights, blood pressures, serum lipid levels, and adipose tissue metabolism of the

mice were comparable between them (Supplementary Fig. 2C–E and Supplementary Fig. 3). The endothelial expression of proinflammatory E-selectin, VCAM1, and ICAM1 were increased in the ligated common carotid arteries from the *Ddr1*WT mice at 1-week post-ligation (before the development of visible plaques), indicative of endothelial inflammation; these increases were suppressed by *Ddr1* depletion (Fig. 1I). Taken together, these results indicate that endothelial DDR1 is required for the endothelial responses to fluid shear stress and the site-specific distribution of atherosclerosis.

### Shear stress induces DDR1 activation and droplet formation

Inspired by the requirement of DDR1 in the EC responses to shear flow, we explored whether DDR1 itself responds to flow. In general, DDR1 is activated through ligand- or ECM stiffness-induced oligomerization, clustering, and phosphorylation[14,25,26]. Immunofluorescence of DDR1 in HUVECs subjected to PS or OS for different time periods showed a shear-induced DDR1 droplet formation (Fig. 2A), echoing our previous findings of vascular smooth muscle liquid-liquid phase separation (LLPS) of DDR1 responding to stiffness[16]. The expression of DDR1 remained unaffected by shear stress (Supplementary Fig. 4). Intriguingly, the formation of DDR1 condensate droplets is rapid and transient (markedly descended over a period of 3 h) in cells subjected to the atheroprotective PS, but slow and sustained in cells under the atherogenic OS (Fig. 2A, B). Assessment of the oligomeric state of DDR1 by using electrophoretic separation under a non-reducing condition indicated that PS rapidly increased the ratios of dimerized and oligomerized DDR1 (e.g., at 1 h), but this increase was indeed induced by OS over time (e.g., at 3-hour) (Fig. 2C, D), in consistent with the observation of droplet formation (Fig. 2A, B). Upon activation, DDR1 oligomerization mediates the autophosphorylation of multiple tyrosine residues within the intracellular JM region and tyrosine kinase domain, including tyrosine-792 (Tyr792)[25]. Analysis of DDR1 Tyr792 phosphorylation under a reducing condition revealed that the phosphorylation level of DDR1 in cells exposed to PS is higher than that to OS for 1 h, and that this differentially regulation was reversed after shearing for 3 or 6 h (Fig. 2E). To observe the shear-induced DDR1 droplet formation in real-time, we used a three-dimensional microfluidic device to produce laminar or disturbed flow by mimicking regions of living blood vessels with 90% lumen obstruction[27] (Fig. 2F). In living HUVECs infected with DDR1-EGFP adenovirus at the laminar-flow region, the shear-induced DDR1 droplets formed on the apical side of the cells, rather than on the basal side (Fig. 2G, Supplementary Movie 1 and Supplementary Movie 2); DDR1 droplets formed and increased rapidly within a few seconds upon shearing but then declined over time (Fig. 2H). Whereas the droplet formation and increase at the disturbed-flow region were delayed to up to 3 h (Fig. 2H), in accordance with the results from the parallel-plate flow apparatus. It is worth noting that while most of the DDR1 under static conditions resided in lipid raft areas, as shown by the lipid raft marker caveolin-1, in reaction to shear stress, only a restricted portion of DDR1 droplets coexisted with the early endosome marker Rab5A, or with Lyso-Tracker Red-labeled lysosomes, or the exocytotic vesicle marker CD63 (Supplementary Fig. 5). This demonstrates the complex interplay between DDR1 droplets and membranous organelles within the cell. To assess the flow-induced DDR1 activation further in vivo, we looked into the endothelial distribution of DDR1 in mouse aortic arch versus thoracic aorta, where disturbed or laminar flow naturally occurs[3]. *En face* immunofluorescence showed pronounced DDR1 condensates appeared in the VE-cadherin-marked ECs in the inner curvature of aortic arch; in contrast, the DDR1 condensates were almost absent in ECs in the thoracic aorta (Fig. 2I, J), suggestive of a flow-pattern-specific activation of DDR1. Regarding the potential interaction between DDR1 and VE-cadherin, the Pearson's correlation coefficient indicated minimal colocalization between DDR1 and VE-cadherin in aortic arch or thoracic aorta (Fig. 2K). The results from live cell imaging,

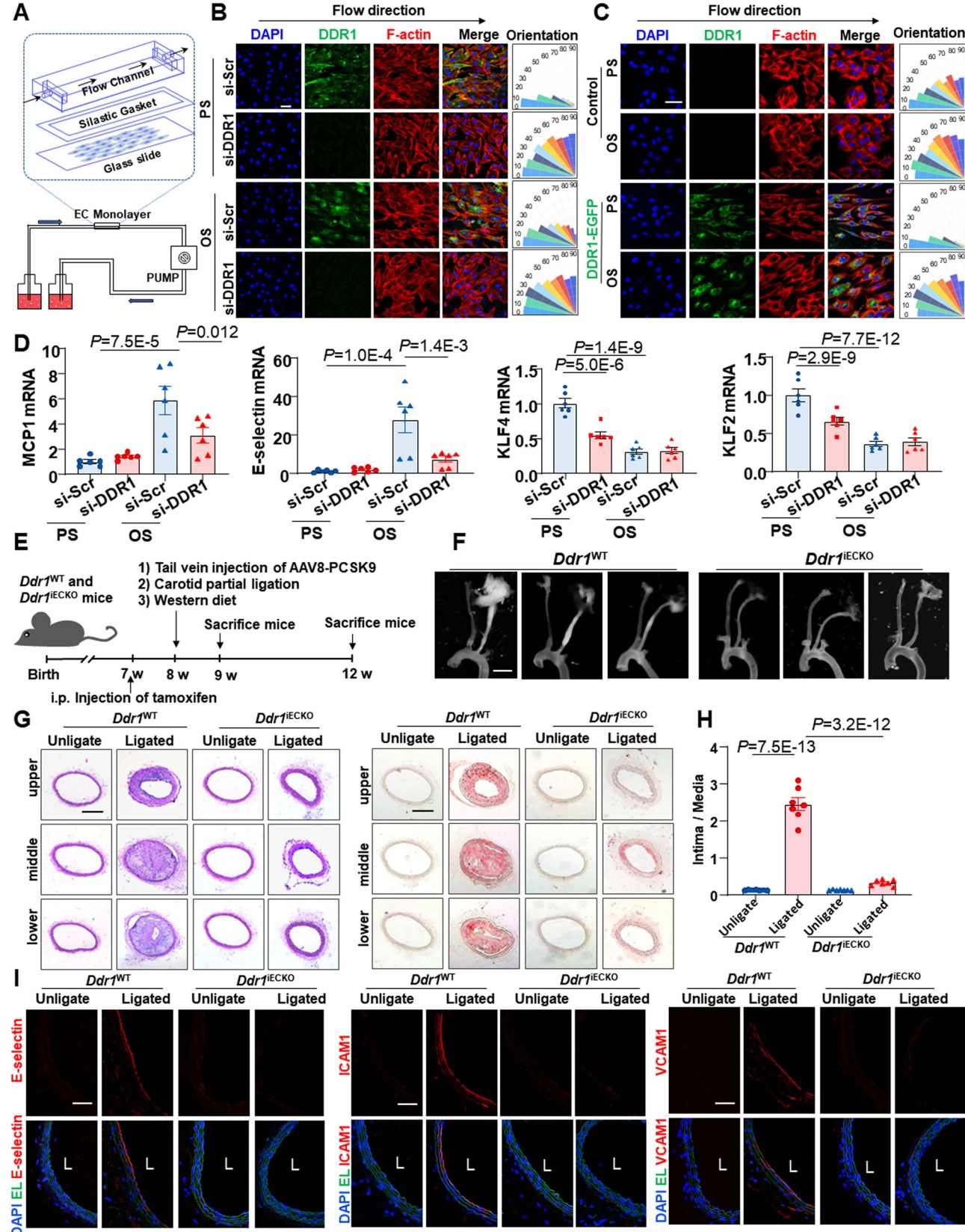

immunofluorescence staining, and co-immunoprecipitation further confirmed the absence of interaction between DDR1 and VE-cadherin in HUVECs (Supplementary Fig. 6). The findings collectively demonstrate a spatial and temporal regulation of DDR1 activation by fluid shear stress.

Additionally, the ability of DDR1 in responding to mechanical forces was confirmed by supporting evidence from HUVECs subjected to various kinds of mechanical stimuli including ECM micropatterning, ECM stiffness, and cyclic stretch. DDR1 droplets were observed in cells in a pathophysiological status, e.g., in a square shape, on stiffer gels, or

**Fig. 1 | DDR1 mediates the endothelial cell responses to fluid shear stress.**
**A** Schematic diagram of parallel-plate flow chamber. **B** Left: Immunofluorescence of F-actin and DDR1 in HUVECs that were transfected with DDR1-siRNA or scrambled control siRNA (si-Scr) and subjected to PS or OS for 24 h. HUVECs were all grown on gelatin-coated glass slide. Right: Quantification of alignment by measuring the orientation angle. $n = 9$ images from 3 biological replicates. Scale bar: 50 μm. **C** Left: Immunofluorescence of F-actin and DDR1 in NIH3T3 cells that were infected with DDR1-EGFP adenovirus and subjected to PS or OS for 24 h. Right: Quantification of alignment by measuring the orientation angle. $n = 9$ images from 3 biological replicates. Scale bar: 50 μm. **D** Quantitative RT-PCR (qRT-PCR) to detect anti-inflammatory transcription factors (KLF2 and KLF4) and proinflammatory genes (MCP1 and E-selectin) in HUVECs treated as in (**B**). Data were analyzed by two-way ANOVA followed by Tukey's multiply test. **E** Schematic

diagram of experimental design. **F** Representative gross images of carotid arteries from the indicated mice at 4 weeks after partial ligation. Scale bar: 2 mm. 3 different positions of carotid artery are showed, and the definitions of upper, middle, and lower are consistent with anatomical position. **G** Representative H&E and Oil red O staining of neointima in the left common carotid arteries from the $Ddr1^{WT}$ and $Ddr1^{iECKO}$ mice at 4wk after ligation. Scale bar: 200 μm. **H** Quantification of the atherosclerotic lesion area. $n = 7$ mice. The most severe atherosclerotic lesion of the left common carotid artery was counted. Data were analyzed by two-way ANOVA followed by Tukey's multiply test. **I** Representative Immunofluorescence staining of E-selectin, ICAM1, and VCAM1 in cross-sections of the left common carotid arteries from the $Ddr1^{WT}$ and $Ddr1^{iECKO}$ mice at 1 week post-operation. Scale bar: 50 μm. Data were expressed as the means ± SEM.

exposed to cyclic stretch with 15% deformation (1 Hz), however, absent in cells in a physiological status, e.g., in an elongated shape, on softer gels, or kept in a static condition (Supplementary Fig. 7).

## DDR1 is a mechanosensor in ECs

We sought to explore whether DDR1 is a primary mechanosensor and capable of detecting mechanical forces directly. We used magnetic tweezers to apply tensional forces (~ 4 pN) to paramagnetic beads coated with DDR1 ligand or antibody that specifically recognizes DDR1 (Fig. 3A), and then examined the force responses in terms of DDR1 clustering/condensation and increase in intracellular calcium levels. In HUVECs expressing DDR1-EGFP, applying forces with magnetic tweezers to the collagen- or DDR1 antibody-coated beads induced noticeable formation of DDR1 clusters surrounding the paramagnetic beads, and this phenomenon was absent if the beads were coated with the control bovine serum albumin or IgG (Fig. 3B, C). The results are in line with the observations by employing the microfluidic device (Fig. 2G, H). Given the indicative role of calcium influx in ECs with respect to activation of tyrosine kinase receptors[28], we examined the intracellular calcium levels in ECs when forces were applied to DDR1. Excitingly, we observed robust and transient increases of intracellular calcium levels with forces applied locally (Fig. 3D, E), further reinforcing the notion of directly sensing forces by DDR1. One proposed mechanism for the shear stress activation of mechanosensors, including the mechanosensitive ion channel Piezo1, involves the mediatory role of plasma membrane tension[29,30]. To explore whether the activation of DDR1 is related to changes in membrane tension, we visualized the EC membrane tension by using a membrane-bound tension biosensor named MSS[31], based on the fluorescence resonance energy transfer (FRET) technique. MSS consists of a tension sensor module, comprising an elastic spider silk protein inserted between ECFP and YPet, and two anchoring proteins linked with lipid molecules in lipid raft and non-lipid raft regions (Fig. 3F). A lower FRET ratio indicates a higher membrane tension. EA.hy926 ECs co-expressing MSS and DDR1-mCherry were subjected to laminar shear for 60 s and then kept in a static condition for another 5 min. Shear flow immediately decreased the FRET ratio and meanwhile, induced the formation of DDR1 condensate droplets. The process is reversible, as membrane tension and the number of DDR1 condensates both gradually returned to the basal levels over time in a static condition (Fig. 3G, H). Furthermore, exposure of cells expressing MSS to a hypotonic stimulus to elevate membrane tension by adding 2 times volume of $H_2O$ into the medium also promoted the DDR1 droplet formation (Supplementary Fig. 8A). Cell membrane tension is reported to be positively related to cell membrane fluidity[31]. To determine whether the shear-induced DDR1 activation is mediated by membrane tension, cells co-expressing MSS and DDR1-mCherry were incubated with methyl-β-cyclodextrin (MβCD)-cholesterol to decrease membrane fluidity[32] and then subjected to shear flow or a hypotonic stimulus. The shear- or osmotic pressure-induced increases of the DDR1 condensate number and size, as well as the increases in membrane tension, were remarkably

inhibited by addition of cholesterol to the membrane (Fig. 3I, J, Supplementary Fig. 8A–C). Assessment of the DDR1 phosphorylation level revealed that decreasing membrane fluidity/tension suppressed the shear-induced DDR1 phosphorylation (Fig. 3K). Moreover, we showed that DDR1 activation by force is independent of Piezo1. This was evident as force-induced DDR1 droplet formation persisted with Piezo1 knockdown, and shear stress-induced DDR1 droplets remained unaffected by Piezo1 knockdown (Supplementary Fig. 9). These data collectively demonstrate that DDR1 is a direct mechanosensor.

## A loose conformation of the DS-like domain is required for DDR1 mechanosensation

To dig into the structural bases of DDR1 responding to mechanical forces, we used single-molecule magnetic tweezers to investigate the unfolding dynamics of DDR1 ectodomain (Fig. 4A). A biotinylated recombinant DDR1 protein (21 ~ 416 amino acids, aa) linked to a modified 2-kilobases DNA handles was anchored by one end with dual digoxin to a slide surface coated with anti-digoxigenin, and the other end was anchored to a streptavidin-coated magnetic bead. The unfolding of DDR1 (21 ~ 416 aa) was induced by the increasing forces (1 ~ 30 pN). Stretching DDR1 resulted in a clear unfolding event as revealed in the force-extension curves, with a force of 20.7 pN (Fig. 4B, C). Subsequently, the anchored DDR1 proteins were subjected to shear flow for 5 min. The similar stretching experiments were performed on the same DDR1 molecules indicated that the unfolding forces decreased to 5.5 pN and that the extension step sizes (an average size of 47.9 nm and 48.0 nm, respectively) were comparable between the before-shear and post-shear (Fig. 4B, C). The distinct decrease in the unfolding force suggests that the sheared DDR1 obtained much lower mechanical stability than the DDR1 without shearing. The DS (28-187aa) and DS-like (188-367aa) domains of DDR1 ectodomain have a similar structure, as both fold into an eight-stranded β-barrel[25]. In the DS domain, the N- and C-terminus are held together by a conserved disulphide bond, while in the DS-like domain, two deeply buried cysteines (Cys303 and Cys348) form an internal disulphide bond[25]. Given these structural characteristics, we estimated the complete unfolding length of the DS domain ($159aa \times 0.36 nm/aa \times 1/2 - 5.0 nm = 23.62 nm$, where 0.36 nm/aa is the length of an aa residue and 5.0 nm is the rough distance between the N- and C-terminus of the DS domain) and the complete unfolding length of the DS-like domain ($(179-23) aa \times 0.36 nm/aa - 6.0 nm = 50.16 nm$, where 23 aa is half of Cys303 to Cys348 and 6.0 nm is the rough distance between the N- and C-terminus of the DS-like domain). Apparently, the unfolding step size measured by magnetic tweezers matched the estimated complete unfolding length of the DS-like domain, suggesting that shear flow may destabilize and loosen the DS-like domain.

Despite that DDR1 and DDR2 share a high degree of sequence identity in these extracellular globular DS (59% identity) and DS-like (51% identity) domains[17], crystal structuring of the ectodomains of DDR1 and DDR2 highlights disparities in the DS-like domain[17,33]. To determine the potential role of the DDR1 DS-like domain in mediating

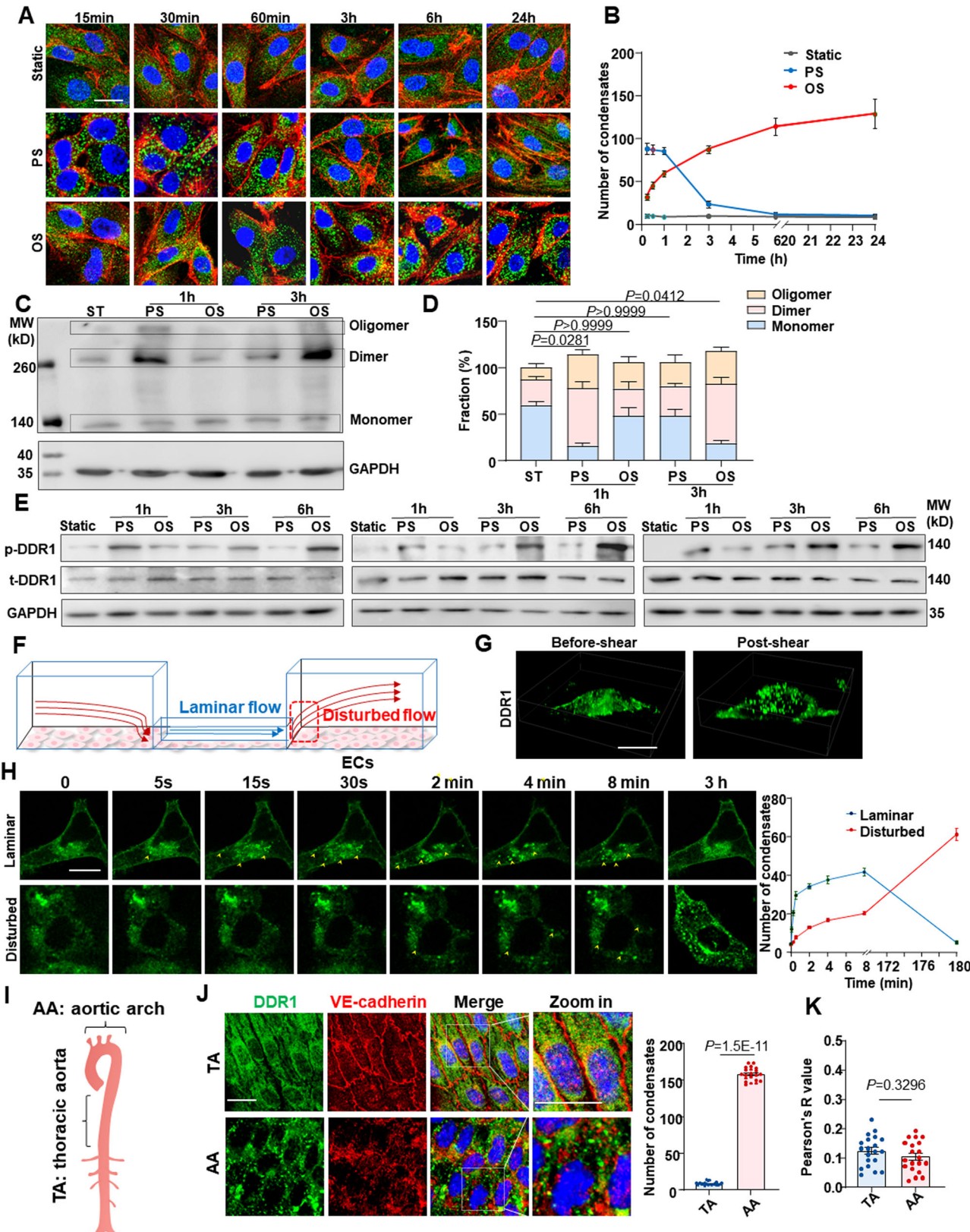

DDR1 activation, we generated a DDR2-EGFP construct and two recombinant mutants of DDR1-EGFP named reDS-like (the DS-like domain in DDR1-EGFP was replaced by the DDR2 DS-like domain) and reDS (the DS domain in DDR1-EGFP was replaced by that the DDR2 DS domain) (Supplementary Fig. 10A). Interestingly, the shear-induced droplet formation was still present in cells expressing reDS, but absent in DDR2-EGFP or reDS-like (Supplementary Fig. 10B). This disparity in their DS-like domains could potentially account for the unresponsiveness of DDR2 to shear stress. In comparison, the DDRs ligand, collagen, could stimulate droplet formation in cells expressing reDS-like and reDS but not in cells expressing DDR2-EGFP (Supplementary Fig. 10C). These results indicate that the DS-like domain is required for

**Fig. 2 | Shear stress induces DDR1 activation and formation of liquid-like biomolecular condensates. A** Representative immunofluorescence of DDR1 in HUVECs subjected to PS or OS for six time periods (15 min, 30 min, 1 h, 3 h, 6 h and 24 h). HUVECs were grown on gelatin-coated glass slide. **B** Quantification of DDR1 condensates in (**A**), $n = 60$ images from 6 biological replicates. Data were expressed as the means ± SD. **C** Non-reducing SDS-PAGE to detect DDR1 oligomerization in HUVECs subjected to PS or OS for indicated time. **D** Quantification of the fraction of DDR1 monomer, dimer and oligomer, which was normalized to GAPDH and performed using ImageJ based on the analysis of the gray band intensity. $n = 4$ biological replicates. Data were expressed as the means ± SEM and analyzed by Kruskal-Wallis test with Dunn's test. *P* value, ratio of DDR1 dimers and polymers to total DDR1. **E** Western blotting to assess DDR1 phosphorylation in HUVECs subjected to PS or OS for indicated time. 3 biological replicates were presented. **F** Schematic diagram of 3D microfluidic vascular model to produce disturbed or laminar flow.

**G** Three-dimensional imaging of DDR1 droplets in HUVECs responding to laminar flow for 2 min. **H** Left: time-lapse images of HUVECs infected with DDR1-EGFP adenovirus, which seeded on gelatin-coated microfluidic chamber and subjected to laminar flow or disturbed flow for 3 h. DDR1 droplets were indicated by yellow arrows. Right: quantification of DDR1 condensates. $n = 6$ biological replicates. Data were expressed as the means ± SEM. **I** Schematic cartoon of aortic arch and thoracic aorta. **J** Left: En-face staining of DDR1 and VE-cadherin in the aortic arch and thoracic aorta of C57BL/6 wild-type mice. Right: quantification of DDR1 condensates. $n = 20$ images from 5 mice. **K** Colocalization analysis of DDR1 and VE-cadherin in (**J**). $n = 20$ images from 5 mice. Pearson's *R* value (above threshold) was calculated by ImageJ Fiji software (Analyze-colocalization-coloc2). In (**J**, **K**), Data were all expressed as the means ± SEM analyzed by two-sided Mann–Whitney test. Scale bars, 20 μm.

the shear-specific activation of DDR1 but is unlikely necessary for the ligand-dependent activation of DDR1. Given the conformational differences between the before-shear and post-shear DDR1 DS-like domain, we speculated that stabilizing the DS-like domain may blockade DDR1 mechanosensation. To test this, we generated a DDR1 DS-like locked mutant, in which the N-terminal aspartic acid-189 and the C-terminal phenylalanine-364 were simultaneously mutated to cysteines (D189C/F364C) to allow a disulphide bond formed to hold the N- and C-terminus (Fig. 4D). We verified the stabilizing effect of "lock" by using magnetic tweezers and found no extension step appeared, indicative of block of the unfolding event (Fig. 4E). Live cell fluorescence revealed that compared with the wild type (WT) DDR1, the locked DDR1 did not form droplets to react to shear and showed markedly compromised droplet formation ability in responding to collagen (Fig. 4F, G), demonstrating that a loose confirmation of the DS-like domain is indeed crucial for DDR1 activation. The DS-like domain of DDR1 contains a third deeply buried cysteine: Cys287 (Fig. 4D)[25]. As the formation of disulfide bonds between cysteine residues is important for DDR1 di- and oligomerization[34], we speculated that destabilizing the DS-like domain may expose the buried Cys287 to facilitate DDR1 activation. To look into this, we created a Cys287-to-alanine (C287A) mutant and tested its responses to shear or collagen. The C287A mutant partially inhibited shear/collagen-induced DDR1 droplet formation (Fig. 4H, I), probably because that Cys287 belongs to a patch of amino acid residues that are crucial for DDR1 activation. To further prove that destabilizing the DS-like domain improves oligomerization of DDR1, we used a dual-color fluorescence recovery after photobleaching (FRAP) assay[35], where mCherry or EGFP was fused to the extra- and intracellular part of DDR1, respectively, and antibody targeting mCherry was applied to immobilize a certain fraction of DDR1, followed by measurements of lateral mobility of the non-immobilized DDR1-EGFP by FRAP. In anti-mCherry polyclonal antibody- incubated cells expressing the WT DDR1-EGFP or locked DDR1-EGFP alone, the mobility of intracellularly tagged WT DDR1 (80.1% ± 3.5%) was comparable with that of the locked DDR1 (79.6% ± 4.9%). In antibody-incubated cells expressing the WT DDR1-EGFP and mCherry-WT DDR1 simultaneously, the WT DDR1-EGFP exhibited only marginal recovery (49.2 ± 9.6% recovery ratio at 120-second after bleaching) of fluorescent signals into the bleached regions (Fig. 4J), indicative of a robust di- or oligomerization that may hinder the mobility of nonimmobilized WT DDR1. In contrast, in cells expressing the locked DDR1-EGFP and mCherry-locked DDR1 simultaneously, the locked DDR1-EGFP exhibited a more remarkable recovery (67.6% ± 4.0%) after bleaching with antibody incubation compared with WT DDR1-EGFP (Fig. 4K), suggestive of a weak di- or oligomerization. We further detected the dimeric and higher oligomeric forms of DDR1 by using electrophoretic separation under a non-reducing condition. The shear-induced DDR1 di- or oligomerization was significantly inhibited by mutation of the WT DDR1 to the locked DDR1 (Fig. 4L). The above findings collectively support a

notion that a loose confirmation of the DS-like domain is required for the DDR1 responses to shear in terms of oligomerization and LLPS (Fig. 4M).

## Shear stress induces the association of YWHAE with DDR1 to regulate DDR1 LLPS

To identify the signaling pathways downstream of DDR1 activation, we analyzed the endogenous DDR1-interacting proteins by using co-immunoprecipitation (co-IP) and mass spectrometry in HUVECs kept in static culture or exposed to PS or OS for 3 h (Supplementary Fig. 11A). 64 proteins were selected with a fold-change of over 1.25 or less than 0.75 in OS *vs.* PS (Supplementary Table 1). Notably, among them is HSPB1, a heat-shock chaperone protein recognized for its role in sustaining the liquid-like characteristics of cytoplasmic proteins capable of undergoing phase separation[36]. They were submitted to the PRISM web server (http://cosbi.ku.edu.tr/prism/index.php) for molecular docking assessment. 17 proteins were predicted to be physically interacted with DDR1 (Supplementary Table 2). Protein-protein interaction analysis by the STRING web server (https://string-db.org) further indicated that the 14-3-3 family proteins (YWHAE and YWHAQ) were the key nodes of the interaction network formed by the identified 17 proteins and DDR1 (Fig. 5A). As mass spectrometry failed to detect YWHAQ in OS or static culture, we did not stay focused on YWHAQ. Co-IP assay in HUVECs verified the OS-enhanced interaction of DDR1 with YWHAE (also named 14-3-3ε) (Fig. 5B). Given a documented role of 14-3-3 proteins in regulating LLPS of their binding partners[37], we assessed the potential influence of YWHAE on LLPS of DDR1. An optoDroplet (Opto-) system[38] was utilized to study the potential LLPS propensity of YWHAE itself, where a chimeric protein was assembled from the full-length YWHAE combined with the light-sensitive oligomerization domain of cryptochrome (CRY2) photolyase homology region to control LLPS. In NIH3T3 cells transfected with Opto-YWHAE, YWHAE alone did neither form droplets nor show increases in droplet number upon blue light stimulation (Fig. 5C). Turbidity (OD600) measurements and microscope observations in the presence of PEG8000 did not detect LLPS of the recombinant YWHAE protein in the tested conditions in cell-free (Fig. 5D). In comparison, formation of YWHAE- and DDR1-colocalized droplets and increases in droplet number upon blue light were observed in cells co-expressing Opto-YWHAE and DDR1-EGFP (Fig. 5E). These findings suggest that YWHAE may co-condensate and form droplets with DDR1. To investigate their co-condensation in a cell-free environment, we attempted to study DDR1 and YWHAE but were unable to purify the full-length DDR1 protein. Given the necessity of the transmembrane domain of DDR1 in driving DDR1 LLPS[16], we created the truncated mutants DDR1-C (416 ~ 913 aa, containing the transmembrane and cytosolic domains) and DDR1-N (21 ~ 443 aa, containing the extracellular and transmembrane domains) (Fig. 5F). The purified DDR1 mutants were incubated separately with varying concentrations of YWHAE. While the average size and number of DDR1-N droplets remained constant in response to YWHAE, the size

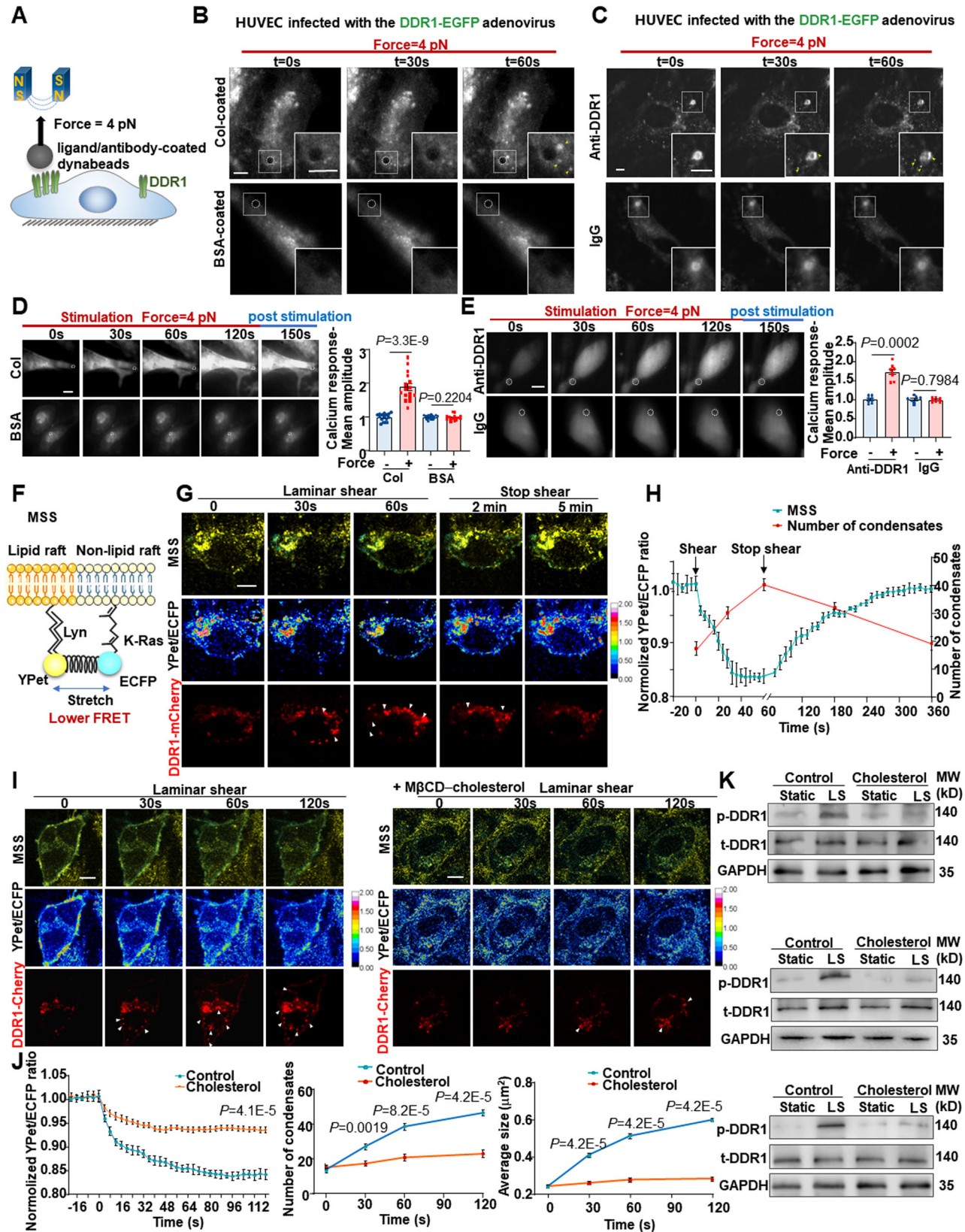

and number of DDR1-C droplets increased with increasing YWHAE concentration, reaching a peak at 5 µmol/L (YWHAE/DDR1 molar ratio=1:1) before decreasing (Fig. 5G). The turbidity assay further confirmed the regulation of DDR1-C LLPS by YWHAE (Fig. 5H). Moreover, FRAP experiments were conducted to investigate whether YWHAE impacts DDR1-C droplet dynamics. The mobility of DDR1

molecules was not affected by the presence of YWHAE in droplets, as FRAP kinetics were consistent across all droplets formed at different YWHAE concentrations (Fig. 5I). To understand the types of interactions between molecules, NaCl and 1,6-Hexanediol (1,6-HD) were used to disrupt electrostatic and hydrophobic interactions, respectively. The addition of 1,6-HD markedly reduced the droplet size and number,

**Fig. 3 | DDR1 is a mechanosensor in endothelial cells. A** Schematic diagram of the magnetic tweezers assay. **B, C** Images of DDR1 condensates formation (yellow arrows) captured by total internal reflection fluorescence microscopy. HUVECs were infected with DDR1 adenovirus and incubated with collagen/ BSA-coated beads in (**B**), and incubated with anti-DDR1/IgG-coated beads in (**C**). Representative images of HUVECs loaded with Fluo-4AM dye and then incubated with collagen/ BSA-coated beads in (**D**), and anti-DDR1/IgG-coated beads in (**E**). Calcium responses were measured by calculating the fluorescent intensity of individual cells before (10 s), during (120 s), and after (30 s) stimulation. The position of dynabeads is indicated by white dashed circles. $n = 16$ cells from 4 biological replicates in (**D**). $n = 8$ cells from 4 biological replicates in (**E**). **F** Schematic diagram of a membrane-bound tension biosensor (MSS). MSS consists of a tension sensor module, which comprising an elastic spider silk protein inserted between ECFP and YPet, and two anchoring proteins. **G** The representative living cell images of YPet/ECFP emission ratio and DDR1-Cherry in EA.hy926 ECs subjected to laminar shear stress for 1 min and post-shear for 5 min. DDR1 droplets were indicated by white arrows. **H** The average time courses of FRET biosensors and quantification of number of DDR1 condensates in EA.hy926 ECs exposed to laminar shear stress. $n = 9$ biological replicates. **I** The representative living cell images of YPet/ECFP emission ratio and DDR1-Cherry in EA.hy926 ECs pretreated without or with cholesterol in complex with methyl-β-cyclodextrin (MβCD) (+MβCD-cholesterol). DDR1 droplets were indicated by white arrows. **J** Left: the average time courses of FRET biosensors in (**I**). Middle: quantification of number of DDR1 condensates in (**I**). Right: quantification of average condensate size in (**I**). $n = 9$ biological replicates. In (**D, E** and **J**), Data were all analyzed by two-sided Mann–Whitney test. **K** Western blotting to assess DDR1 phosphorylation in HUVECs subjected to laminar shear stress for 1 h or static. 3 biological replicates were presented. Data were all expressed as the means ± SEM. In (**B, C**), Scale bars, 5 μm. In (**D, E, G** and **I**), Scale bars, 10 μm.

as well as the turbidity of both DDR1-C and the DDR1-C-YWHAE mixture. The reductions were more pronounced with the addition of NaCl (Fig. 5J, K), indicating that DDR1 LLPS is likely driven by both electrostatic and hydrophobic interactions, while the interaction between DDR1 and YWHAE is mainly mediated by electrostatic interactions. To identify the critical binding sites between DDR1 and YWHAE, we analyzed the predicted interaction interfaces using PRISM and found that residues 670 to 693 were crucial for their interaction (Supplementary Fig. 11B). Thus, we created two mutant forms of DDR1, DDR1 (1 ~ 669 aa) and DDR1 (1 ~ 693 aa). Immunofluorescent imaging showed that DDR1 (1 ~ 669 aa) failed to colocalize with YWHAE, while DDR1 (1 ~ 693 aa) and the full-length version of DDR1 exhibited strong colocalization with YWHAE (Fig. 6A). Further analysis revealed that lysine-674 (K674) and serine-677 (S677) of DDR1 and tyrosine-214 (Y214) of YWHAE might be important, so we mutated these residues to alanine (A) to created K674A and S677A for DDR1, and Y214A for YWHAE. DDR1-K674A (but not DDR1-S677A) and YWHAE-Y214A interrupted the colocalization of DDR1 with YWHAE (Fig. 6B). The PyMol server visualized the structural details of the DDR1-YWHAE interface (Fig. 6C). Co-IP assay confirmed that DDR1-K674A disrupted the interaction between DDR1 and YWHAE (Fig. 6D). Altogether, these findings demonstrate that DDR1 directly binds to YWHAE via its C-terminal intracellular domain and that the co-condensation of DDR1 and YWHAE is dependent on specific residues.

### DDR1-YWHAE is required for the shear stress-induced activation of YAP

The mechanoresponsive YAP is phosphorylated and inhibited by tumor suppressors LATS1/2, and YAP phosphorylation on serine-127 (S127) results in its association with 14-3-3, leading to cytoplasmic localization and degradation of YAP[39,40]. Given the significance of 14-3-3 in retaining YAP in the cytoplasm, we tested whether the DDR1-YWHAE (14-3-3ε) axis modulates the YAP signaling. Exposure of HUVECs to OS for 24 h induced the nuclear translocation and S127 dephosphorylation of YAP and increased the expression of YAP-targeted genes CTGF, CYR61 and ANKRD1 (Fig. 7A–C), consistent with the prior research[19,20]. We found that using DDR1-IN-1, an inhibitor that binds to DDR1's kinase domain and inhibits autophosphorylation, effectively reduced the observed changes (Fig. 7A–C). However, it's noteworthy that despite treatment with OS and DDR1-IN-1, cells still displayed DDR1 droplets. This persistence can likely be attributed to the crucial role of the DDR1 transmembrane domain in driving DDR1 LLPS, as previously demonstrated in our research[16]. To further explore the role of YWHAE in this regulation, we conducted siRNA knockdown experiments. Sole knockdown of YWHAE significantly enhanced YAP nuclear translocation in the presence of PS conditions. However, in the context of OS, the impact of si-YWHAE on YAP seemed less significant (Supplementary Fig. 12). These discrepancies could potentially be attributed to the role of YWHAE suppression in mediating the OS-induced YAP signaling. Moreover, si-YWHAE reversed the inhibitory

effects of DDR1-IN-1 (or knockdown of DDR1) on YAP nuclear translocation, S127 dephosphorylation, and the expression of YAP-targeted genes (Fig. 7D–G). These results collectively demonstrate that the DDR1-YWHAE axis is required for the activation of YAP induced by shear stress (Fig. 8).

Our previous research has demonstrated physical interaction between DDR1 and LATS1[16]. Here, we observed colocalization of DDR1 and LATS1 in HUVECs subjected to PS and OS and that the factors characterizing this colocalization exhibited fluctuations across distinct time frames of shear stress (Supplementary Fig. 13A, B). In an effort to elucidate the role of the DDR1-LATS1 axis in mediating OS-induced YAP activation, we conducted experiments using a combination of DDR1 inhibitor and LATS inhibitor. Immunostaining results revealed that LATS inhibition using Lats-IN-1 could partially mitigated the inhibitory effect of DDR1-IN-1 on YAP activation (Supplementary Fig. 13C, D). This insight underscores the partial involvement of the DDR1-LATS1 interaction in the regulatory process of YAP by DDR1. It is important to note, however, that our focus on the DDR1-LATS1 interaction has been relatively limited, given that LATS1 did not emerge in our mass spectrometry results.

## Discussion

For decades, it has been recognized that the endothelium has a key role in mediating the responses of blood vessels to homodynamic shear stress[2]. However, knowing exactly how these cells sense blood flow to trigger mechanotransduction and functional regulation has been difficult to understand. Mechanosensors are responsible for detecting forces and the initiation of the signal transduction cascade. To be categorized as a mechanosensor, a protein or structure must meet several important criteria[7,41]. It must be expressed in specific cells, essential for the immediate signaling response of cells to the relevant force, activated by the relevant mechanical force when expressed in heterologous cells, and activated independently of biochemical cues or other mechanosensors. Here, we provide evidence that DDR1, which belongs to a new class of receptor-type tyrosine kinases, satisfies all the criteria for being categorized as a mechanosensor. Firstly, our analysis using immunofluorescence and Western blotting, as well as quantitative PCR, showed that endogenous expression of DDR1 is present in vascular ECs of both human and murine origin. Secondly, loss-of-function studies revealed that DDR1 is essential in the orientation of ECs induced by shear stress and impacts the regulation of mechanoresponsive genes. Furthermore, we conducted direct mechanical loading studies using magnetic tweezers to apply a minimal tensional force on the DDR1 present on the EC surfaces that resulted in calcium influx. This is crucial as forces at a small scale, such as 4 pN, do not interfere with the plasma membrane or other mechanosensors[42]. Thirdly, we overexpressed human DDR1 correspondingly into NIH3T3 fibroblast cells-cells that display no expression of DDR1, causing cell orientation in response to physiologically relevant kinds of flow stimulations. We further monitored and observed

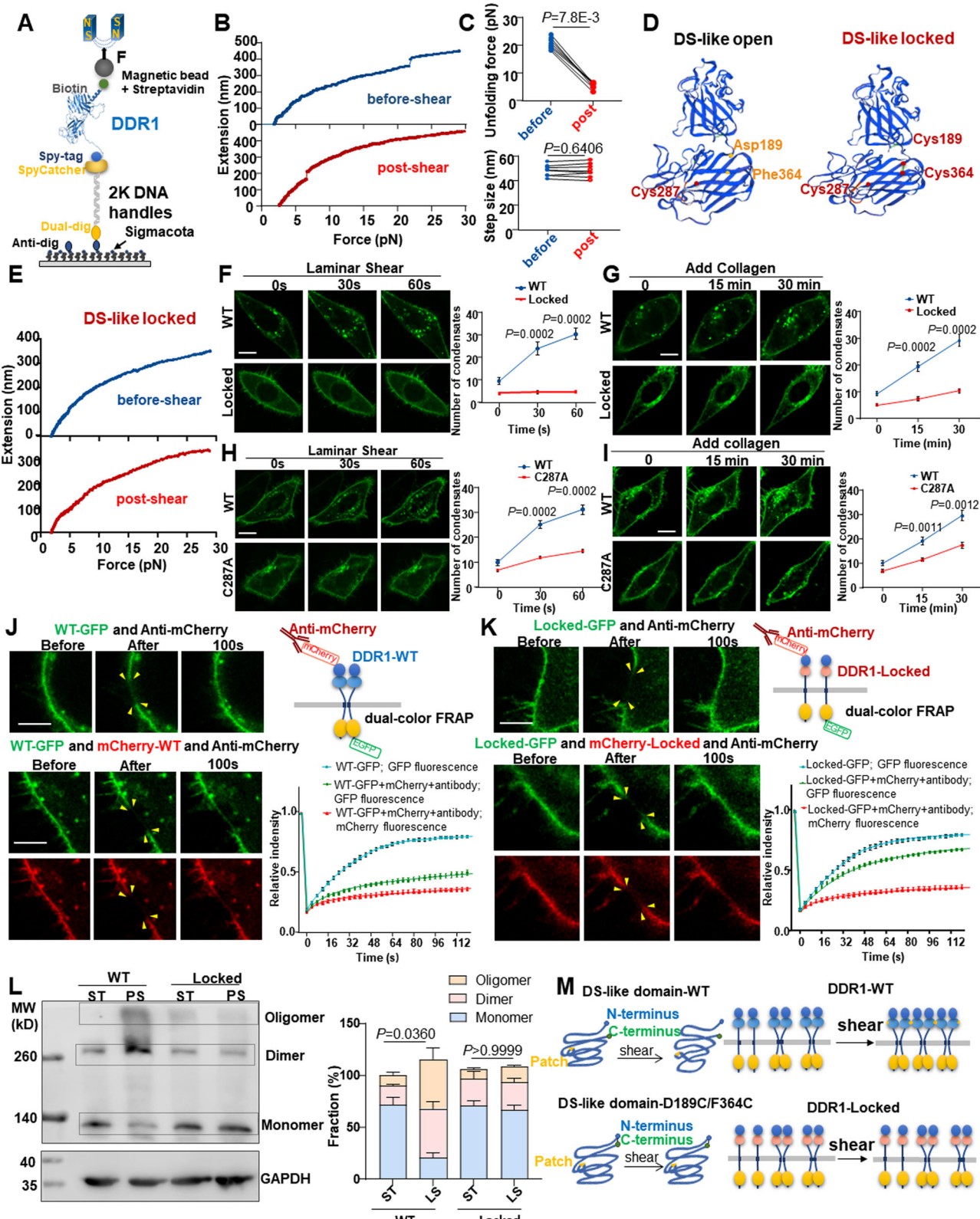

the exogenously expressed fluorescent fusion DDR1 forming droplets in reaction to tensional force or shear stress using live-cell imaging. Lastly, our experiments, in which cells were seeded on gelatin-coated surfaces and subjected to tensional forces directly applied to DDR1, showed unique mechanical activation of DDR1 that was independent of its ligand or other mechanosensory machineries. How does shear stress activate DDR1? Our research utilizing a FRET-based membrane-

bound tension biosensor presented data showing that the activation of DDR1 induced by shear force in terms of the formation of DDR1 droplets and the increase of droplet number was at least partly mediated by plasma membrane tension. This mechanism has been previously reported in other known mechanosensors such as the mechanosensitive ion channel Piezo1[11,43] and mechanosensitive G-protein coupled receptors (GPCRs)[44]. How does DDR1 activation trigger

**Fig. 4 | The open conformation of DS-like domain is crucial for DDR1 mechanosensation. A** Schematic diagram of single-molecule magnetic tweezers measurements. **B** The representative force–extension curve of DDR1 extracellular domain protein pretreated without or with shear for 5 min. **C** Quantification of unfolding force and extension step size of DDR1 extracellular domain protein before- or post-shear. $n = 7$ biological replicates. Data were analyzed by Wilcoxon matched-pairs signed rank test (two-tailed). **D** Left: DDR1 extracellular domain crystal structure generated by SWISS-MODEL. Right: the expected structure of DS-like domain-locked mutant (D189C/F364C). **E** The representative force–extension curve of DDR1 locked mutant (D189C/F364C) pretreated without or with shear for 5 min. **F** Time-lapse images and quantification of DDR1 condensates in ECs transfected with DDR1 WT or locked mutant. **G** Live cell imaging and quantification of DDR1 WT or locked mutant in ECs after 10 µg/ml soluble collagen I stimulation. **H** Time-lapse images and quantification of DDR1 condensates in ECs transfected with DDR1 WT or C287A mutant. **I** Live cell imaging and quantification of DDR1 WT or locked mutant in ECs after 10 µg/ml soluble collagen I stimulation. In (**F–I**), data

were expressed as the means ± SEM and analyzed by two-tailed Mann–Whitney test. $n = 8$ biological replicates. **J, K** Left: dual-color FRAP of EA.hy926 ECs expressing DDR1 (WT/locked)-GFP and mCherry-DDR1 (WT/locked). ECs were subjected to 5 min pre-shear before FRET. The photobleaching area was indicated by yellow arrows. Right: schematic of mCherry-DDR1 (WT/locked) and DDR1 (WT/locked)-GFP expressed in ECs treated with anti-mCherry, and quantification of fluorescent intensity of the photobleaching area. $n = 24$ cells from 6 biological replicates. **L** Left: non-reducing SDS-PAGE to detect DDR1 oligomerization in EA.hy926 ECs transfected with DDR1-WT or DDR1-D189C/F364C(locked), ECs were subjected to PS for 1 h or static. Right: Quantification of the fraction of DDR1 monomer, dimer and oligomer. $n = 4$ biological replicates. Data were expressed as the means ± SEM and analyzed by Kruskal-Wallis test with Dunn's test. $P$ value, ratio of DDR1 dimers and polymers to total DDR1. **M** Schematic diagram of DS-like domain structure and DDR1 oligomerization before- or post-shear. Data were all expressed as the means ± SEM. In (**F–I**), Scale bars, 10 µm. In (**J, K**), Scale bars, 5 µm.

calcium influx? A previous study demonstrated that tyrosine kinase receptor activation prompts the entry of extracellular calcium in bovine aortic ECs, possibly achieved by activating calcium-permeable channels through receptor interactions[28]. Phosphorylation of tyrosine kinase receptors on their tyrosine residues was also shown to regulate calcium entry[45,46]. Therefore, it is plausible that DDR1 activation induces calcium influx through receptor phosphorylation, subsequently activating calcium-permeable channels.

The upregulation of proinflammatory molecules (MCP1, ICAM1, and E-selectin) and downregulation of anti-inflammatory transcription factors (KLF2 and KLF4) induced by disturbed flow has been closely linked to an atherogenic endothelium phenotype that is susceptible to atherosclerosis[47]. Our research identified DDR1 as a vital mediator of endothelial responses to shear stress, and our findings showed that endothelial-specific depletion of DDR1 could counteract proinflammatory responses in ECs and abrogate atheroma formation in areas of disturbed flow. Although Bendeck and colleagues previously reported the pro-atherogenic role of DDR1 in atherosclerosis by using systemic *Ddr1* depletion mice, they mainly attributed the effect to the deficiency of DDR1 in vascular smooth muscle cells that led to changes in matrix accumulation and plaque composition[48,49]. The pro-atherogenic role of DDR1 was also supported by their findings demonstrating that the selective deletion of DDR1 in bone marrow cells led to a marked decrease in atherosclerosis[50]. As for a role of DDR1 in ECs, it has been found to mediate high glucose-induced EC senescence and lymphangiogenesis[51,52]. However, our study is the first to report the role of DDR1 in regulating endothelial responses to fluid shear stress and the resulting atherosclerosis. The role of DDR1 in physiological laminar shear stress responses in vivo has yet to be determined, but it will be worthwhile to investigate in future studies.

Experiments conducted in the 1980s established a paradigm for activation of tyrosine kinase receptors: ligand binding leads to receptor dimerization, which in turn facilitates the autophosphorylation of critical tyrosine residues located in the activation loop of the catalytic kinase domains, ultimately leading to kinase activation[53]. DDR1 exhibits several unusual characteristics that distinguish its activation from that of typical tyrosine kinase receptors. DDR1 forms stable noncovalent dimers in the absence of ligand and the process of ligand-induced DDR dimerization seems to involve multiple interactions in the extracellular, transmembrane, and cytosolic regions, with the transmembrane region being the key dimerization interface[54]. Upon binding to collagen via its extracellular DS-domain, DDR1 undergoes oligomerization, aggregation/clustering, autophosphorylation, kinase activation, and endocytosis[17,18,55]. DDR1 dimerization has been linked to the leucine zipper motif in the transmembrane domain[54]. Additionally, research has shown that the 199 ~ 412 aa residues spanning the DS-like domain and extracellular JM region are also necessary for DDR1 dimerization[34]. However, there is limited understanding of how DDR

oligomerization occurs and the specific role that the DS-like domain plays in DDR1 di- or oligomerization. A prior study using antibodies to block the epitopes located on the DS-like domain effectively stabilized the inactive confirmation of DDR1[56], suggestive of the importance of the DS-like domain. Our current study has provided further insights by showing that the DS-like domain undergoes conformational changes in response to shear force, which lead to the exposure of buried C287 that can facilitate disulphide bond formation between DDR1 monomers or dimers. To support this mechanism, we have generated a "locked" mutant to stabilize the DS-like domain and a C287A mutant to prevent disulphide bond formation. Our results demonstrate that the DS-like domain plays a crucial role in DDR1 di- or oligomerization, as the locked mutant exhibited severe impairment in receptor di- or oligomerization and response to shear force, while the C287A mutant showed milder impairment. Our findings shed light on the mechanism by which the DS-like domain mediates DDR1 di- or oligomerization and ultimately leads to DDR1 activation.

LLPS in cellular physiology is commonly viewed as a mechanism of organizing molecules amidst a crowded and disordered environment, achieved through the reversible concentration of specific molecules and the exclusion of others[57]. It compartmentalizes proteins and nucleic acids into micron-scale, liquid-like, membraneless biomolecular condensates with specific functions[58], including RNA metabolism, ribosome biogenesis, the DNA damage response and signal transduction[59]. Studies have reported that the main kinases involved in the Hippo pathway (LATS1/2), along with their signaling effectors (YAP and TAZ), can undergo phase separation and interact with other proteins within condensed droplets[60–62]. Despite functioning as a chaperone-like protein that binds to YAP to regulates its degradation, 14-3-3 has not been observed to undergo LLPS alone or co-condense with YAP. However, the 14-3-3 zeta isoform (14-3-3ζ) has been found to participate in the LLPS of other proteins, such as Tau[63], suggesting that 14-3-3 may play a regulatory role in LLPS. Our prior study has demonstrated that DDR1 is able to undergo LLPS in response to ligand binding or matrix stiffening[16]. How does the LLPS of DDR1 in ECs affect the shear-activation of YAP? We identify 14-3-3ε (YWHAE) as a regulator of DDR1 LLPS that can participate in the signaling cascade initiated by DDR1 activation and lead to the nuclear translocation of YAP. YWHAE forms co-condensates with DDR1, increasing the size and number of droplets and insulating itself from association with YAP to mediate cytoplasmic retention of YAP. Our data establish a correlation between the LLPS of mechanosensory proteins and the activation of the YAP pathway.

## Methods
### Animals
All studies were performed in accordance with the guidelines of the Animal Care and Use Committee of Peking University and approved by

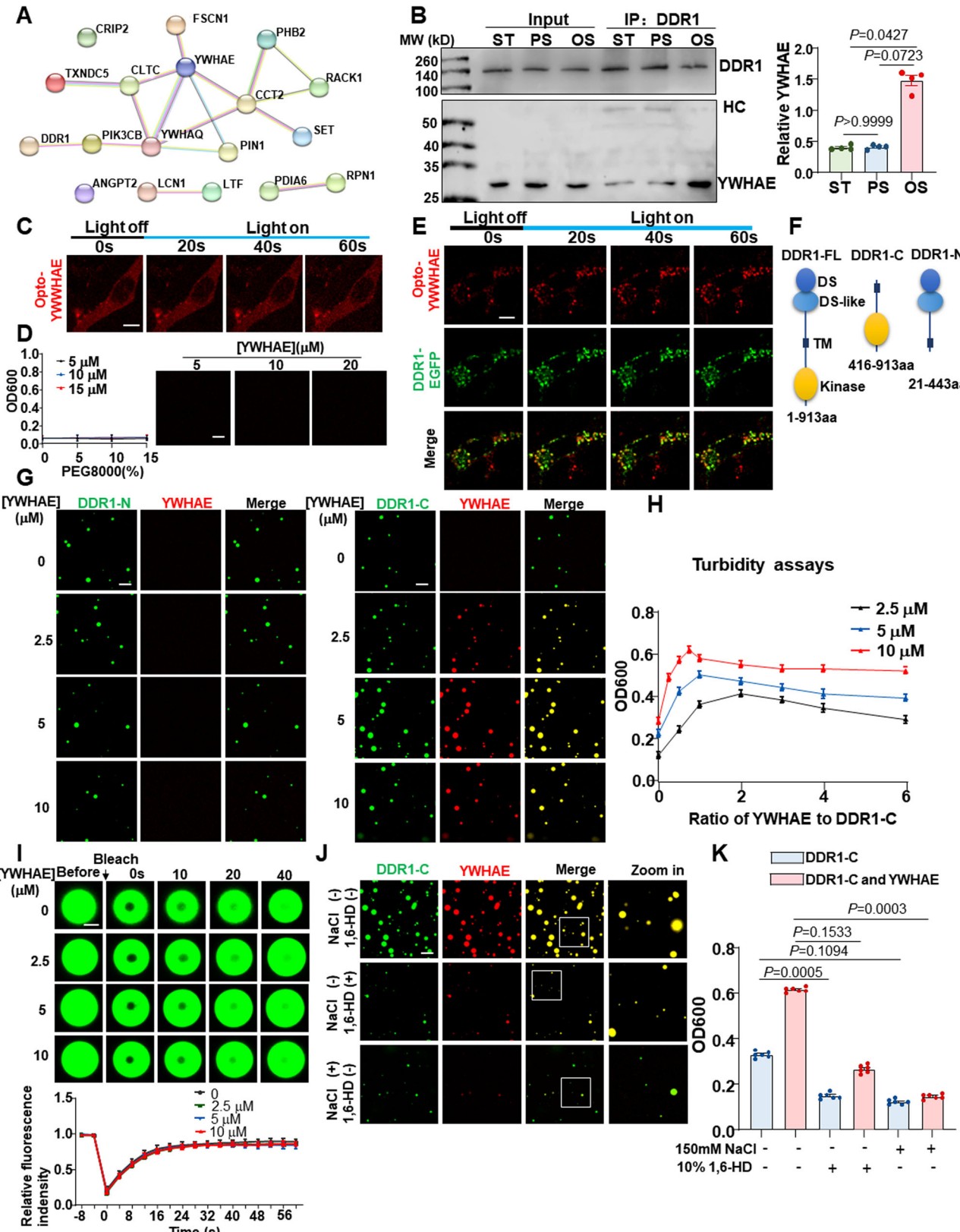

the Ethics Committee of Peking University Health Science Center (LA2015017). All Experimental protocols were approved by Peking University Health Science Center. Wild-type mice (C57BL6J, 8-10 weeks old) were obtained from the Experimental Animal Center at Peking University Health Science Center (Beijing, China) and housed under controlled conditions with 23 ± 1 °C of ambient temperature and 60%

of relative humidity. They were synchronized to a 12 h light-dark cycles with free access to food and water. In order to generate mice with conditional knockdown of Ddr1 specific to ECs, we employed the *CreER^T2*-loxP-mediated recombination system. Mice carrying the Ddr1 coding region flanked by loxP sites (*Ddr1* ^flox/flox^) were generated by Cyagen Biosciences (China). *Ddr1* ^flox/flox^ mice were maintained in a

**Fig. 5 | Shear stress induces the association of YWHAE with DDR1 to regulate DDR1 phase separation. A** Protein-Protein interaction analysis of the IP-MS identified proteins. **B** Left: Co-IP between DDR1 and YWHAE in HUVECs cultured under the static condition or subjected to PS or OS for 3 h. IP was performed with anti-DDR1 antibody. HC: Heavy chain. Right: quantification of YWHAE immunoprecipitated with DDR1. $n = 4$ biological replicates. Data were expressed as the means ± SEM and analyzed by Kruskal−Wallis test with Dunn's test. **C** Representative images of NIH3T3 cells expressing Opto-YWHAE. Scale bar, 10 μm. **D** Left: turbidity of YWHAE solution at different concentrations. Data were expressed as the means ± SD. $n = 5$ biological replicates. Right: representative fluorescence microscopy images of YWHAE solution at different concentrations. Scale bar, 10 μm.
**E** Representative images of NIH3T3 cells expressing both Opto-YWHAE and DDR1-EGFP. Scale bar, 10 μm. **F** Schematic representations of full-length DDR1 (DDR1-FL), 416-913aa of DDR1 (DDR1-C) and 21-443aa of DDR1 (DDR1-N). **G** Cell-free phase

separation assay showing droplet formation of YWHAE-Cherry with DDR1 (416-913aa)-EGFP (DDR1-C, 5 μM) or DDR1 (21-443aa)-EGFP (DDR1-N, 5 μM) in indicated concentrations. Scale bar, 10 μm. In (**C**, **E**, and **G**), experiments were repeated 3 times independently with similar results. **H** Turbidity assays of LLPS of DDR1 (416-913aa)-EGFP at different concentrations and YWHAE/DDR1 molar ratio. Data were expressed as the means ± SEM. $n = 5$ biological replicates. **I** Up: representative fluorescence microscopy images of DDR1 droplets with different YWHAE concentrations. Down: quantification of fluorescent intensity of the area indicated by the dashed circle after FRAP experiment. $n = 6$ biological replicates. Data were expressed as the means ± SD. Scale bar, 1 μm. **J** Representative fluorescence microscopy images of droplets formed by DDR1 with YWHAE in the presence of 1,6-HD or 150 mM NaCl. Scale bar, 10 μm. **K** Turbidity of DDR1 and DDR1/YWHAE in the presence of 1,6-HD or 150 mM NaCl. $n = 6$ biological replicates. Data were expressed as the means ± SEM and analyzed by Kruskal−Wallis test with Dunn's test.

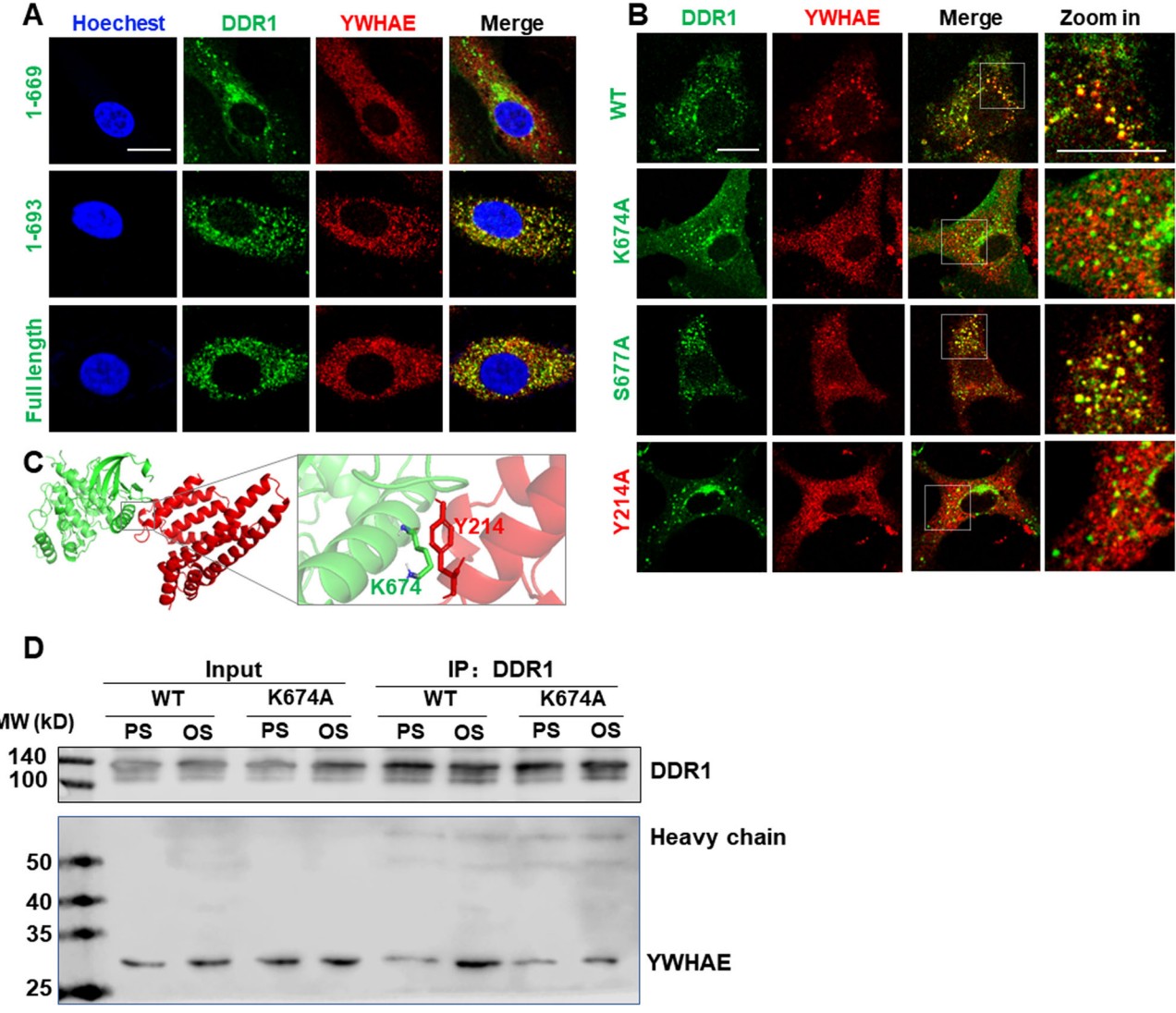

**Fig. 6 | DDR1 can interact with YWHAE by C-terminal domain. A** Live cell imaging of DDR1(1 - 669 aa), DDR1(1 - 693 aa), DDR1-FL and YWHAE-Cherry in EA.hy926 ECs. ECs were subjected to disturbed flow for 3 h in microfluidic chambers. **B** Live cell imaging of DDR1-WT/K674A/S677A and YWHAE-WT/Y214A in EA.hy926 ECs. ECs were subjected to disturbed flow for 3 h. **C** Structural details of the DDR1-YWHAE interfaces generated with PyMol, DDR1 and YWHAE residues are shown as green and red sticks, respectively. **D** Co-IP between DDR1 and YWHAE in EA.hy926 ECs subjected to PS or OS for 3 h. Cells were transfected with the indicated constructs. IP was performed with anti-DDR1 antibody. Scale bars, 20 μm. In (**A**, **B**, **D**), experiments were repeated 3 times independently with similar results.

C57BL/6 background and crossed with vascular endothelial-cadherin Cre recombinase-positive mice (*Cdh5-CreER^T2*), to generate the *Ddr1*^flox/flox^ *Cdh5-CreER^T2+^* (*Ddr1*^iECKO^) mice. *Ddr1*^flox/flox^ *Cdh5-CreER^T2-^* (*Ddr1*^WT^) littermates were used as the controls. 4 male mice and 3 female mice

were included in each group of partial carotid ligation. Primers for genotyping are listed in Table S6. Mice were subjected to tamoxifen (dissolved in corn oil) intraperitoneal injection with 50 mg/kg every day for 7 days.

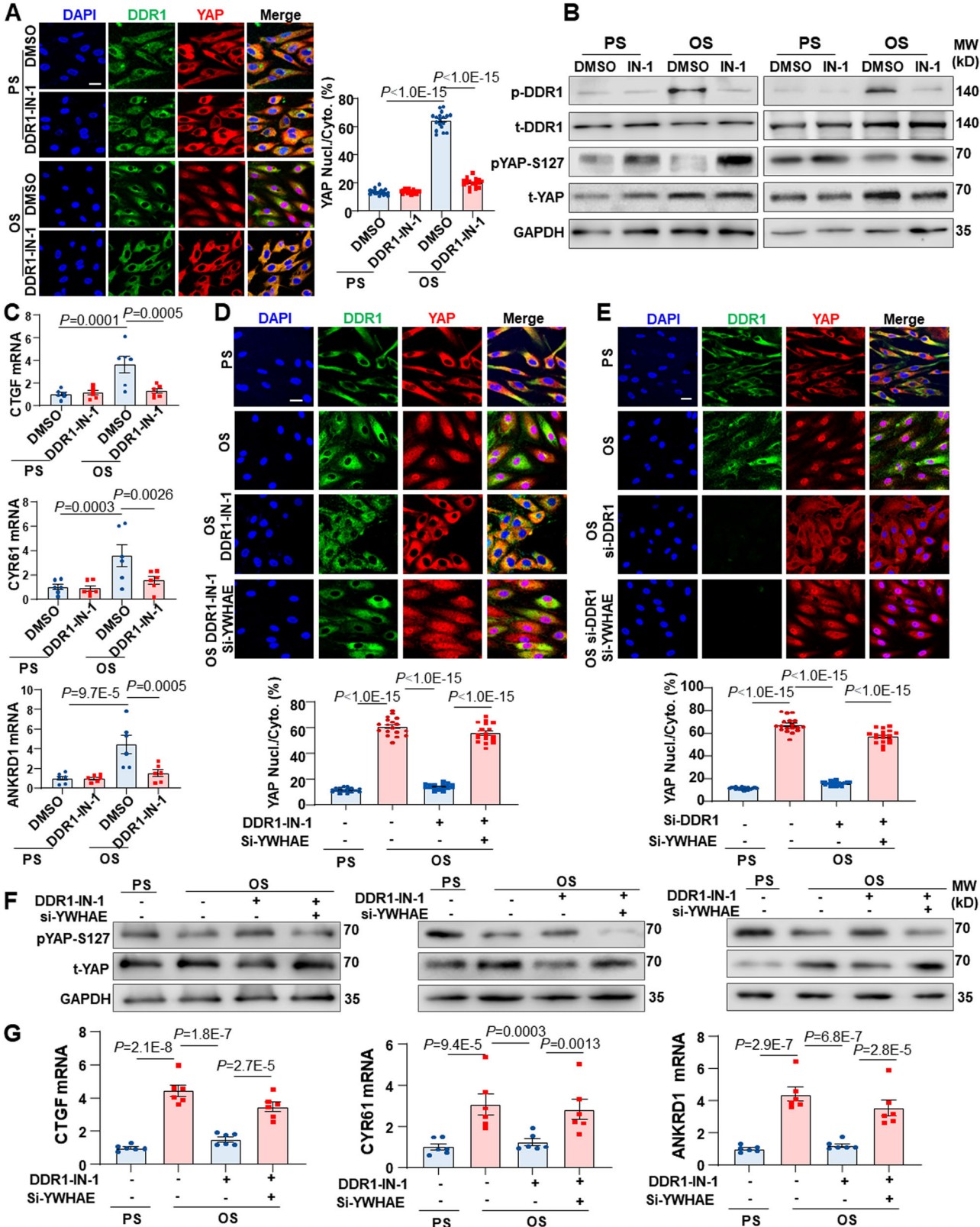

## Cell culture

HUVECs were isolated from umbilical cords from healthy patients after full-term deliveries. The gender of the fetus includes both males and females. De-identified umbilical cords were obtained with the agreement of the patients and approved by the Peking University People's Hospital Medical Ethics Committee (2015PHB024). During the application, endothelial cells from 3–5 umbilical cords are mixed together. HUVECs within passages 5–8 were maintained in Medium 199 supplemented with 10% fetal bovine serum (FBS) (900-008; Gemini), 4 g/mL of endothelial cell growth factor (ECGF) (E1388; Sigma) at 37 °C in an incubator with 95% humidified air and 5% $CO_2$ and passaged every 3 days. EA.hy926 (CL-0272) and NIH3T3 cells (CL-0171) were obtained from Pricella Biotechnology (Wuhan, China) and authenticated using STR (short

**Fig. 7 | DDR1-YWHAE is required for the shear stress-induced activation of YAP.**
**A** Immunofluorescence and quantification of YAP localization in HUVECs. The cells were subjected to PS or OS for 24 h. $n = 18$ images from 3 biological replicates. 10 - 15 cells per image. **B** Western blotting to assess DDR1 and YAP phosphorylation in HUVECs subjected to PS or OS for 24 h. 3 biological replicates were presented. **C** Quantitative RT-PCR analysis of the expressions of YAP target genes CTGF, CYR61 and ANKRD1 in HUVECs treated as in (**A**). $n = 6$ biological replicates. **D** Immunofluorescence and quantification of YAP localization in HUVECs subjected to PS or OS for 24 h. The cells were transfected with siRNAs specific for YWHAE or scrambled siRNA and incubated with DDR1-IN-1 (10 μmol/L) or the control reagent

(DMSO) for 3 h before being subjected to shear. $n = 18$ images from 3 biological replicates. 8 - 12 cells per image. **E** Immunofluorescence and quantification of YAP localization in HUVECs subjected to PS or OS for 24 h. The cells were transfected with si-DDR1, both si-DDR1 and si-YWHAE, or scrambled siRNA. $n = 18$ images from 3 biological replicates. 12 - 16 cells per image. **F** Western blotting to assess DDR1 and YAP phosphorylation in HUVECs treated as in (**D**). 3 biological replicates were presented. **G** Quantitative RT-PCR analysis of the expressions of YAP target genes CTGF, CYR61 and ANKRD1 in HUVECs treated as in (**D**). $n = 6$ biological replicates. Data were all analyzed by two-way ANOVA followed by Tukey's multiply test. Scale bars, 20 μm. Data were all expressed as the means ± SEM.

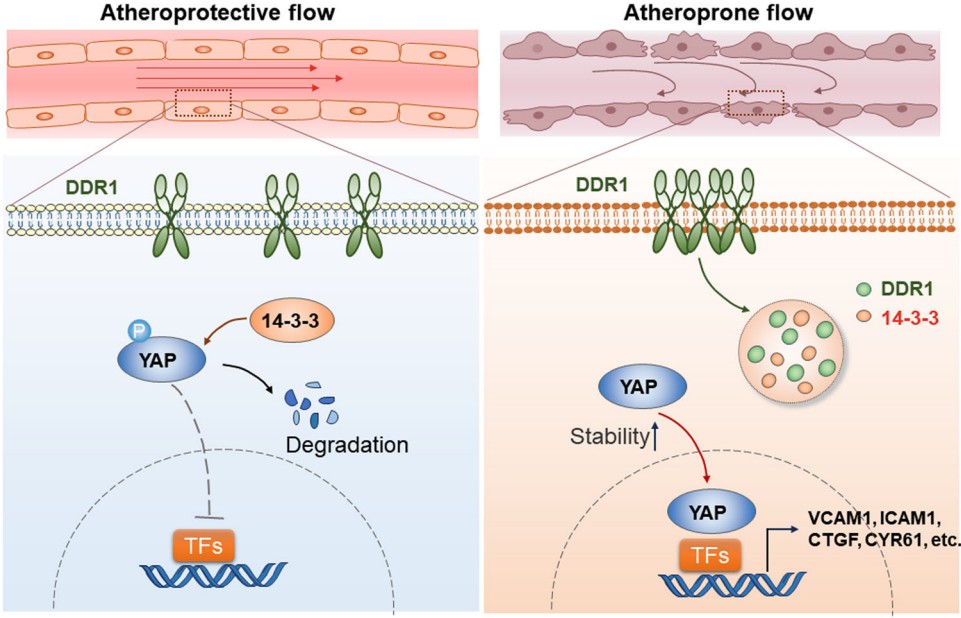

**Fig. 8 | DDR1 mechanosensation mediates the flow-activation of YAP and endothelial dysfunction.** The schematic depicts DDR1 as a primary mechanosensor in ECs, orchestrating cellular responses to shear flow. In areas of atheroprotective laminar flow, YAP phosphorylation prompts its binding with 14-3-3 proteins. This leads to YAP localization in the cytoplasm and subsequent degradation. Conversely, in atheroprone disturbed flow areas, DDR1 perceives the flow, initiating force-induced DDR1 oligomerization. This results in the formation of liquid-like biomolecular condensates involving DDR1 and 14-3-3. These condensates inhibit YAP phosphorylation and cytoplasmic sequestration, resulting in YAP activation and consequent endothelial dysfunction.

tandem repeats) testing. EA.hy926 and NIH3T3 cells were cultured in Dulbecco Modified Eagle's Medium (CT0035001; HARVEYBIO) containing 10% FBS (900-008; Gemini).

#### Flow experiments

Monocultured HUVECs seeded on 0.1% gelatin (MB1720; HARVEY-BIO)-coated glass slides were subjected to shear stress in a parallel-plate flow apparatus. The flow channel in the chamber was created by a silicon gasket with dimensions of 2.5 cm in width (w), 5.0 cm in length, and 0.025 cm in height (h). The chamber containing the cell-seeded glass slide fastened with the gasket was connected to a perfusion loop system, kept in a constant-temperature controlled enclosure, with pH maintained at 7.4 by continuous gassing with a humidified mixture of 5% (vol/vol) $CO_2$, 20% (vol/vol) $O_2$, and 75% (vol/vol) $N_2$. The shear stress (τ) generated on the HUVECs was estimated as 6Qμ/wh2, where Q is flow rate, w is dimension in width and μ is perfusate viscosity. The flow of pulsatile shear (PS, 12 ± 4 dynes/cm2) or oscillatory shear (OS, 0.5 ± 4 dynes/cm2) is composed of mean flow with shear stress at 12 dynes/cm2 or 0.5 dynes/cm2 supplied by a sinusoidal oscillation using a piston pump with a frequency of 1 Hz and a peak-to-peak amplitude of ± 4 dynes/cm2. HUVECs were perfused in M199 medium containing 2% FBS and NIH3T3 cells were perfused Dulbecco Modified Eagle's Medium containing 2% FBS in for the indicated time.

#### Analysis of cell orientation

To analyze cell orientation, we employed the OrientationJ plugin within the Fiji software to determine the orientation of actin filaments. In a netshell, the image data was initially subjected to skeletonization using a Hessian-based multiscale filter, followed by a conversion into a binary label matrix. Subsequently, structural elements were fitted to the skeleton, and segments smaller than 5 pixels were disregarded. The maximum Feret diameter of each segmented filament was identified as the longest possible distance between two parallel tangents at opposing structural element borders. The angle between the maximum Feret diameter relative to the direction of applied flow was then calculated. The colored ray plots serve to illustrate the cumulative distribution of filaments within specific angle ranges relative to the flow direction, categorized as 0–10, 11–20, 21–30, 31–40, 41–50, 51–60, 61–70, 71–80 or 81–90, respectively.

#### Microfluidic model

This model was adapted from Ingber et al.[27] Microchannels mimicking vascular constriction were prepared from polydimethylsiloxane (PDMS) using conventional soft lithography. A master mold was prepared by aligning 80 μm layers designed using a computer-aided design program and formed using a cutter plotter (CE5000, Graphtec, CA). The device contained a region (160 μm in height × 400 μm in width × 10 mm in length) with a 90% constriction relative to upstream

and downstream channel regions (each: 640 μm in height × 2 mm in width × 20 mm in length). Within the microfluidic chamber, the application of geometric obstruction leads to a remarkable ≈100-fold amplification in shear stress at the stenotic site. This results in the establishment of distinct shear stress patterns. Specifically, the region characterized by elevated and unidirectional shear stress, situated at the stenotic area, is deemed to be representative of a laminar flow environment. Conversely, just downstream of the narrowed region, a region with lower and reciprocating shear stress prevails, indicative of a disturbed flow area. The PDMS channels were sealed with glass using plasma bonding. The microfluidic devices were sterilized using ethanol and UV irradiation and perfused with gelatin (0.1%) to support cell adhesion. HUVECs were introduced to the microchannel and allowed to adhere under a static condition (4 h at 37 °C). The devices were then placed in a tissue culture incubator. To observe DDR1 activation induced by shear stress in real time, microfluidic chamber was placed in the living cell workstation and M199 medium containing 2% FBS was recirculated through the microfluidic device using a syringe pump (LSP02-2A, LongerPump, China). The flow rate was adjusted to obtain a wall shear stress of 0.5 dyne/cm2 (≈10 μL/min) at the unconstructed channels and laminar shear stress of 10 dyne/cm2 at the constriction region. Images were captured by using Leica SP8 confocal microscope.

## Coverslip cleaning and surface functionalization for single-molecule magnetic tweezers

To anchor the protein to the surface, the bottom coverslip needs to be cleaned and functionalized. Firstly, several pieces of coverslips were put in a staining jar filled with deionized water and detergent, and then the staining jar was placed in an ultrasonic cleaning bath for 30 min. After that, the coverslips were rinsed with deionized water for 5–10 times to remove the detergent. The washing process was repeated using acetone and methanol, respectively to rinse some organic impurities. To functionalize the surface of coverslip, the siliconizing reagent (Sigmacote, SL2) was coated on the surface and heated at 120 °C for 3 min, then the surface could bind with anti-digoxin for tethering modified DNA handle. To control the thermal drift, the coverslips were covered with polystyrene beads (2 μm, QDSpher). These firmly bound beads served as reference beads for drift correction in experiments.

## Single-molecule measurements

The setup of magnetic tweezers is mainly composed of light source (625 nm red LED, THORLABS), two NdFeB magnets with 0.5 mm, flow cell, 40× oil immersed objective (UPLSAPO60XO, NA 1.35, Olympus), and 1280 × 2024 CCD camera (MC1362, Mikrotron). The flow cell consists of a bottom coverslip, a double-sided tap with a rectangular channel (5 × 50 mm2), and an upper coverslip with an inlet and outer at each end. The recombinant DDR1 protein (21-416aa) carrying N-terminal Avi-tag was biotinylated by Avi labelled protein biotin labeling kit (P0630S; Beyotime). Protein was crossed with Oligo-SpyCatcher by C-terminal Spy-tag and then crossed with modified 2 kilobase DNA handles. In the flow cell, one end of recombinant DDR1 protein (21-416aa) is anchored to the streptavidin coated dynabeads M-280 (11205D; Invitrogen), and the other end is anchored to the surface coated with anti-digoxigenin (11214667001; Sigma-Aldrich). The images of beads would go through the objective and then into the CCD camera. The real-time position of the bead at various forces was recorded by comparing the diffraction pattern of the bead with calibration images at various distances from the focal point of the objective. In LabView software, the quadrant-interpolation (QI) algorithm enables highly parallel single-molecule experiments help to trace the dynamics of folding and unfolding at high-throughput level. All measurements were carried out at 25 °C. To explore the mechanical unfolding of DDR1 protein undergo shear stress, 5 min of pre-shear was performed. Pre-shear was completed with a syringe pump (LSP02-2A;

LongerPump), and the shear stress of flow is 0.6 dyne/cm2. The raw data of force-jump assay was processed by using a MATLAB script.

## Bead pulling/magnetic tweezer system

To couple protein/antibodies to magnetic beads, the Dynabeads Antibody Coupling Kit (14311D; Invitrogen) was used. 50ug of protein was coupled to 5 mg of Dynabeads M-270 (2.8 μm, Invitrogen Norway). HUVECs infected with DDR1-EGFP virus were incubated with the beads (coated with DDR1 ligand or antibody) prior to force application for 30 min at 37 °C. An empty glass bottom dish was covered with poly-styrene beads, which served as reference beads for drift correction in each experiment. Image of DDR1-EGFP droplets was acquired with total internal reflection fluorescence microscope (TIRF, Olympus).

## Calcium imaging

HUVECs were cultured in 33 mm glass bottom dishes to form a sub-confluent monolayer. After the cells were fully attached, 4 μM of Fluo-4 AM calcium binding dye (F14217; invitrogen) was added to the media. Cells were incubated with the dye for 30 min. Beads conjugated with either collagen or DDR1 antibody (sc-390268; Santa Cruz) were added and the cells were incubated for another 30 min. To assess the calcium influx as a result of mechanical stimulation, the cells with Fluo-4 and magnetic beads were subjected to a force of 4 pN applied with magnetic tweezer. Time-lapse video of calcium imaging was acquired with the total internal reflection fluorescence (TIRF) microscope for 10 s before stimulation, 120 s stimulation and 30 s after stimulation. Acquired image sequences were analyzed by measuring the mean fluorescence intensity (mean pixel value) of each cell at each frame. Mean peak amplitude was calculated and normalized to the pre-stimulation fluorescence intensity for each cell.

## Polyacrylamide (PA) gel preparation

Gels with different stiffness were obtained by varying the final concentrations of acrylamide (A4058; Sigma-Aldrich) (4% and 8%) and bis-acrylamide (M1533; Sigma) (0.15% and 0.264%) for the corresponding elastic moduli (2.55 and 19.66 kPa), respectively. The mixtures containing acrylamide and bis-acrylamide were degassed for 20 min to remove oxygen from the solutions. To polymerize the mixtures, 4.5 ml of 10% w/v ammonium persulfate (Bio-Rad) and 3 ml of N, N, N9, N9-tetramethylethylenediamine (TEMED; Bio-Rad) were added to yield a final volume of 500 mL PA solution. A 50–70 mm-thick gel was cast, and the thickness was confirmed through imaging of embedded fluorescent beads (Invitrogen) in the gel by confocal microscope. To cross-link extracellular matrix proteins onto the gel surface, the gels were activated by exposing the heterobifunctional crosslinker sulfo-SANPAH (Pierce, 0.5 mg/ml in PBS) to UV 264 nm light and subsequently were coated with 0.1% gelatin (MB1720; HARVEYBIO) for each gel at 37 °C for 1 h.

## Cyclic stretch experiment

The model of cyclic stretch loading on cultured HUVECs was established using the ATMS Boxer Cyclic Stretch Culture System (ATMS-Boxer, Taiwan). Briefly, ECs were seeded on gelatin (MB1720; HAR-VEYBIO)-coated flexible-silicone slides (ATMS #CH2020). The cells in the flexible-silicone were then incubated at 37 °C in a humidified 5% $CO_2$ atmosphere overnight at a density of 70–80% confluence. Subsequently, ECs were subjected to uniaxial cyclic stretch (15% deformation) at 1 Hz. Cells cultured under the static condition were considered as the control group.

## Antibodies

Primary antibodies against pDDR1-Y792(11994S; CST; 1:1000), DDR1 (10536-1-AP; Proteintech; 1:1000), YAP1 (13584-1-AP; Proteintech; 1:1000), pYAP1-S127 (AP0489; ABclonal; 1:1000), YWHAE (11648-2-AP; Proteintech; 1:1000) VE-cadherin (bs-4310R; Bioss; 1:1000) and GAPDH

(BE0024; Easybio; 1:4000) were used for Western blotting. Rhodamine Phalloidin (RM02835; ABclonal; 1:200), Primary antibodies against YAP1 (66900-1-Ig; Proteintech; 1:200), DDR1 (10536-1-AP; Proteintech; 1:200), VE-Cadherin (sc-9989; Santa Cruz; 1:100), E-selectin (MA1-22165; Invitrogen; 1:200), ICAM1 (sc-7891; Santa Cruz; 1:100), VCAM1 (sc-13160; Santa Cruz; 1:100), Caveolin-1 (66067-1-Ig; Proteintech; 1:200), Rab 5A (sc-166600; Santa Cruz; 1:100), PE-conjugated CD63 (BC-353003; BaiCheng Technology; 1:200) and LATS1 (3477S; CST; 1:200) were used for immunofluorescence staining. DDR1 antibody (200 µg/ml; sc-390268; Santa Cruz) were used for bead pulling/magnetic tweezer system. The validation of all primary antibodies for the species and application were described on the manufacturer's website.

## RNA isolation and quantitative RT-PCR

RNA from cultured cells was also extracted by TRIzol reagent according to the manufacturer's instructions. Isolated RNAs were reversed-transcribed into complementary DNA by Hifair III 1st Strand cDNA Synthesis SuperMix (11141ES60; Yeasen). Real-time PCR was performed with the 2×RealStar power SYBR Mixture (11202ES08; Yeasen) by using the specific primer pairs. The thermocycler conditions used in amplification are shown below: denaturation at 95 °C for 10 s, annealing at 55 °C for 20 s and extension at 72 °C for 20 s. All primers are list in Table S3. We used $\Delta\Delta$Ct method for analysis. Gene expressions were normalized against GAPDH.

## Western blotting

Cells were lysed in RIPA buffer (MP015; Macgene), containing 1×PMSF and phosphatase inhibitors. Protein concentration was determined with the Bicinchoninic Acid (BCA) assay by Pierce™ BCA Protein Assay Kit (23250; Thermo Fisher). For nonreducing SDS-PAGE on 6% polyacrylamide gels, samples were resuspended in non-reducing sample buffer (BL529B; Biosharp). 10% resolving gels were used in all experiments except for nonreducing SDS-PAGE. Equal amounts of proteins were separated on sodium dodecyl sulphate-polyacrylamide gel electrophoresis (SDS-PAGE), transferred to nitrocellulose membranes, blocked with 5% bovine serum albumin (BSA) in Tris-buffered saline (TBS) containing 0.1% Tween 20 (TBST), and incubated with the indicated primary antibodies overnight at 4 °C. After incubation with HRP-conjugated goat anti-rabbit/mouse IgG (CW0103S/CW0102S; Procell) diluted 1:4000 in TBST, the blots were visualized by Molecular Imager ChemiDoc XRS + System (Bio-Rad).

## Plasmids and transient transfection

Plasmid containing full-length DDR1b isoform was a kind gift from Dr. Christopher A. McCulloch. DDR1b plasmid was used to generate point mutants K674A, S677A and C287A by Fast Mutagenesis System (FM111-02; TRANS). Plasmid containing Cry2 (CRY2olig-mCherry) was purchased from Addgene (60032) to generate the Opto-droplet Optogenetic system (Opto-YWHAE). Plasmid containing CDH5(human) was purchased from MiaoLingBio (P29785). All plasmids were constructed using ClonExpress Ultra One Step Cloning Kit (C115-01; Vazyme) to ligate PCR products into pcDNA3.1 backbone vector opened by restriction digestion (BamHI and EcoRI). EGFP fragment was subcloned from EGFP-Vimentin-7 (Addgene: 56439), and mCherry fragment was subcloned from CRY2olig-mCherry (Addgene: 60032). All primers used for plasmid construction are listed in Table S4. All generated plasmid sequences were confirmed by DNA sequencing. NIH3T3 Cells at 80% confluence were transfected with designated plasmid using lipofectamine 2000 transfection agent (Thermo Fisher) according to the manufacturer's instructions. HUVECs and EA.hy926 ECs were transfected in suspension using the Lonza nucleofector. For loss-of-function studies of DDR1 or YWHAE, cells at 80% confluence were transfected with siRNAs specific for DDR1, YWHAE, Piezo1 or the scrambled siRNA (Table S5). $10^7$ cells were mixed with 500 µL nucleofection buffer and the pre-defined HUVEC nucleofection protocol on the instrument was used for electroporation of 50 nmol/L siRNA, in agreement with standard nucleofection protocols. After electroporation, the nucleofection buffer with cells was added to fresh culture media of endothelial cells without growth factors.

## Immunofluorescence, live cell imaging, and the quantification

For immunofluorescence, cells were fixed in 4% PFA for 15 min and permeabilized with 0.2% Triton X-100 in PBS for 5 to 10 min. Non-specific binding was blocked by 3% BSA for 30 min. The primary antibody was diluted 1:200 in 3% BSA and incubated overnight at 4 °C, then probed with secondary antibody including Alexa Fluor 488-conjugated goat anti-rabbit/-mouse IgG (ab150077/ab150113; Abcam) or Alexa Fluor 555-conjugated goat anti-rabbit/-mouse IgG (ab150078/ab150114; Abcam) diluted 1:200 in PBS for 1 h at room temperature. Nuclei were counterstained with DAPI. Images in either fixed cells or live cells were captured by using Leica SP8 confocal microscope or stimulated emission depletion (STED) super-resolution microscopy.

For quantification of subcellular distribution, the regions of cell nucleus or cytoplasm were manually determined, and measurements of fluorescence intensity were conducted by using NIH ImageJ software. Statistic data were obtained from approximate 100 - 200 cells from 18 random fields from 3 independent experiments. Each dot represents the mean value of all the cells from one random field.

## Membrane-bound biosensor

The membrane-bound tension biosensor, named MSS, is a kind gift from Dr. Bo Liu. Specifically, MSS is constructed by PCR with a forward primer containing 21 amino acids from Lyn kinase and a reverse primer containing 14 amino acids from K-Ras, which can link the tension sensor to phospholipids in the region of lipid raft and non-lipid raft, respectively.

## Cholesterol treatment

Cholesterol in complex with methyl-β-cyclodextrin (MβCD) was purchased from Sigma. Cells were incubated with cholesterol-enriching agents for 30 min at 37 °C. The final concentration of cholesterol in the medium is 0.1 mmol/L.

## Fluorescence recovery after photobleaching (FRAP) assay

FRAP assay was performed by Leica SP8 confocal microscope. For FRAP of DDR1 droplets in vitro, GFP signals in regions of interest (ROI) were bleached by using a 488 nm laser beam at 80% power. For analysis of receptor oligomerization by FRAP, GFP was excited by the 488 nm laser line and mCherry was excited by the 561 nm laser line. The settings for scanning were scanning at 1000 Hz, image format 512 × 512 pixels. We adjusted laser intensity for photobleaching to obtain 70% to 90% loss of fluorescence in the 3 µm × 1 µm rectangular bleached region in the equatorial plane of the cell membrane. To allow for rapid bleaching, we used a 488 nm laser beam at 100% power for a single bleaching scan. We collected images before and after bleaching using low laser intensities and monitored FRAP for 120 s to 240 s. The fluorescence intensity between pre-bleaching and the time point right after bleaching was recorded by microscope. Data were collected by LAS X microcope software. For FRAP curves fitting, logarithmic equation fitting was used in Microsoft Excel.

## Recombinant proteins

For single-molecule magnetic tweezers experiment, Constructs encoding the ectodomain (residues 21-416) of DDR1 wild-type or D189C/F364C double mutant carrying N-terminal Avi-tag and C-terminal Spy-tag and hexahistidine (His6) tag. Protein was expressed by E. coli BL21 (DE3), purified by HisTrap FF Crude (29048631; Cytiva) and slowly dialyzed with PBS buffer. Avoid adding strong reducing agents such as DTT during protein purification process. YWHAE-Cherry recombinant protein was purchased from Dai-An

Biotechnology (Wuhan, China). A DNA sequence encoding YWHAE (Uniprot No.: P62258) fused with C-terminal mCherry tag was cloned into the pET32a vector. The recombinant YWHAE protein was expressed by E. coli BL21 (DE3), and the purity of protein (>90%) was determined by SDS-PAGE. For DDR1(1-443aa)-EGFP and DDR1 (416-913aa)-EGFP recombinant protein expression, a DNA sequence encoding DDR1(21-443aa)-EGFP or DDR1 (416-913aa) was cloned into pDG02583 (104129, Addgene) vector which opened by restriction digestion (EcoRI and HindIII) with an N-terminal MBP-bdSUMO-tag. The recombinant DDR1 protein was expressed by Transetta (DE3) Chemically Competent Cell (CD801, TransGen Biotech), induced for 16 h by adding 0.2 mmol/L IPTG at 16 °C, purified by Ni-NTA column in buffer containing 20 mmol/L Tris-HCl (pH 7.0), 500 mmol/L NaCl and 10% glycerin, and finally concentrated to 20 mmol/L using Amicon Ultra 4 mL 50 kDa centrifugal filter (Merck).

## Turbidity assay

This assay measures the turbidity of the phase-separated solution by absorbance at 600 nm with NanoDrop (Thermo scientific). Protein samples were prepared by mixing determined amounts of targeted protein with buffer containing 5% PEG8000 to achieve desired concentrations of each component. Absorbance at 600 nm was monitored and recorded at room temperature by NanoDrop. Absorbance values were reported after subtracting the optical density of buffer.

## In vitro droplet assay

Purified proteins were diluted to different concentrations with a buffer containing 20 mM Tris-HCl (pH = 7.0), 75 mmol/L KCl and 1 mmol/L DTT, and the mixtures were incubated at 4 °C for 5–10 min to induce phase separation. 20 μl of the turbid solution was injected into glass-bottom dish and imaged by Leica SP8 confocal microscope.

## Co-immunoprecipitation

Cells were lysed with NETN buffer with 1× protease inhibitor cocktail (Roche) and 1× PMSF for 30 min on ice, followed by centrifugation at 13,000 g for 10 min at 4 °C. Supernatants were then transferred into new Eppendorf tubes and incubated with indicated antibodies or IgG overnight at 4 °C on a ferris wheel. After that, add 20 ul of protein A/G agarose beads to the tubes and incubated 1.5 ~ 2 h at 4 °C. Beads were subsequently washed three times with NETN buffer and boiled with equal volumes of 2×SDS loading buffer.

## Identification of DDR1-binding proteins

Cell lysates of HUVECs were from 3 independent biological replicates, and each group containing 100 ug of proteome sample. 10ul anti-DDR1 monoclonal antibody (Santa Cruz; sc-374618) was added to cell lysates. Pull-down lysates were separated by SDS-PAGE and submitted for mass spectrometry analysis at Quantitative Proteomics Center of Peking University Health Science Center. The original data of mass spectrometry is available in the last sheet of Source data. In the analyzed co-IP samples, the bait, DDR1, was present in nearly equal amounts in different conditions, reflecting the quantitative robustness of both the co-IP conditions in addition to the quality control assessment in terms of equal loading of peptides from each condition. Fold change is calculated based on normalized abundances. Protein-protein interaction was explored using STRING analysis (https://string-db.org/).

## Protein interaction analysis

Protein structure files were downloaded from the UniProt database (https://www.uniprot.org/) and processed by PyMOL 2.4, including the removal of redundant protein chains. The identifier of DDR1 structure that we downloaded was 6BRJ, and the identifiers of other protein structures were listed in Supplementary Table S2. After that, we used the PRISM web server (http://cosbi.ku.edu.tr/prism/index.php) to predict the potential interaction interfaces between two proteins. PyMOL 2.4 was used to visualize the docked conformation.

## Partial carotid ligation

Mice were anesthetized by intraperitoneal injection of 1.25% avertin, and the left carotid bifurcation of mice was exposed following a neck incision. Three branches (external carotid, internal carotid, and occipital) of the left carotid artery were ligated with a 6-0 silk suture, and the superior thyroid artery was left intact. The right carotid arteries were served as controls. Mice were injected once with adeno-associated-virus-8 (AAV8)-overexpressing PCSK9 (AAV8-PCSK9) via the tail vein and fed on a Western diet (D12108C, high-fat rodent diet with 1.25% cholesterol). At 4-week after the operation, 7 mice of each group were sacrificed and the vessels were perfused with a fixative (4% paraformaldehyde in PBS) under a pressure of 100 mmHg. The carotid arteries were harvested to prepare the frozen section.

## En face analysis of aortic endothelium

Mice were anesthetized and fixed with 4% paraformaldehyde in PBS buffer for 10 min by perfusion through left cardiac ventricle under physiological pressure. Aortas were harvested, further fixed with 4% paraformaldehyde in PBS buffer for 20 min and were then longitudinally dissected with microdissecting scissors. The luminal surfaces of the aortas were exposed, blocked with 3% bovine serum albumin in PBS for 1 h at room temperature, and were incubated with primary antibodies against DDR1 (10536-1-AP; Proteintech), VE cadherin (sc-9989; Santa Cruz) at 4 °C overnight. The aortas were washed three times with PBS and were then probed with secondary antibodies including Alexa Fluor 488-conjugated goat anti-rabbit IgG (1:500) or Alexa Fluor 555-conjugated goat anti-mouse IgG (1:500). Nuclei were counterstained with DAPI for 5 min at room temperature. Image acquisition was performed using a laser-scanning confocal microscopy (Leica TCS SP8).

## Tail-cuff measurement of blood pressure and heart rate

Blood pressure and heart rate were recorded by using a CODA Mouse & Rat Tail-Cuff Blood Pressure System (Kent Scientific Co., Connecticut, USA). Mice were placed in the restraint corridor and allowed at least 10 min of acclimation. The area was warmed with a heating pad and a quiet, dark environment was maintained to ensure reliable measurements within the parameters of this technology. The mice underwent 7 consecutive days of training sessions from 3 to 6 PM each day to become accustomed to the tailcuff procedure. 15 cycles per measurement were daily performed on each mouse and the blood pressure was the mean value of all successful measurements.

## Serum biochemical analysis

Blood samples were collected into heparin-coated tubes. The samples were centrifuged at 3,000 × g for 15 min to obtain plasma that was used for determination. Total plasma cholesterol (CHO) (BioSino, 000180), triglycerides (TGs) (BioSino, 000220), low-density lipoprotein cholesterol (LDL-C) (BioSino, 020245), and high-density lipoprotein cholesterol (HDL-C) levels (BioSino, 020235) were determined using assay kits.

## Hematoxylin and eosin staining

After rehydration, 7 μm sections were stained with hematoxylin solution for 20 s followed by 2 dips in 1% acid ethanol (1% HCl in 70% ethanol) and then rinsed in running water. Then the sections were stained with eosin solution for 3 min and followed by dehydration with graded alcohol (70%, 80%, 90%, 95%, 100%) and clearing in xylene. The plaque areas were determined using ImageJ software and calculated by expressing the plaque/neointima area relative to tunica media area.

## Oil red O staining

The mice were sacrificed at 4-week postsurgery and the carotid arteries were harvested to prepare the frozen section. After fixation in 4% formaldehyde for 5 min, the tissues were first rinsed with PBS for 10 min and then rinsed with 60% isopropanol at room temperature. The sections were stained with Oil Red O (0.2% in 60% isopropanol) for 10 min with gentle shaking and rinsed again with 60% isopropanol and then with water for three times and covered with glycerin mounting medium.

## Statistical analysis and reproducibility

We used G*Power 3.1.9.7 to conduct priori power calculations. Analyses were performed by using GraphPad Prism version 8.0.1. The exact $P$-value is provided in the figures. Animals were randomized before assignment to different group or treatment, and randomization was performed by using random number generator. Data collection and analysis were conducted in a blinded way. To achieve blind analysis, two different investigators were involved as follows: the first researcher randomly numbered each group. The second researcher, responsible for the statistical analysis, could only see the number and was blind to treatment assignment. Random numbers were generated using the standard = RAND () function in Microsoft Excel. We used blind analysis in atherosclerotic lesion area analysis and all immunofluorescence imaging analysis. For animal experiments and in-vitro experiments involving a large group of cells, all the data with $n \geq 6$ were tested for normality using an Aderson-Darling, D'Agostino-Pearson, Kolmogorov-Smirnov or Shapiro–Wilk test. For normally distributed data, differences between treatment groups were determined using unpaired $t$-test for two groups of data and one-way or two-way ANOVA for multiple groups of data since the variance is homogeneous. Statistical significance among multiple groups was determined by post hoc analysis (Tukey's honestly significant difference test). Nonparametric tests were used for $n < 6$. The Kruskal–Wallis test was conducted to test statistical significance for experiments with $\geq 3$ groups followed by Dunn's post-hoc test, and the Mann–Whitney test was used to examine statistical significance between 2 groups. For cellular immunofluorescence imaging analysis, unpaired $t$-test for 2 groups of data and one-way ANOVA followed by Tukey's multiply test for 3 groups of data were used when cell number $\geq 30$, and nonparametric tests were used when cell number $< 30$. For single molecule magnetic tweezers experiment, Wilcoxon matched-pairs signed rank test was used. Values of $P < 0.05$ were considered statistically significant. All experiments were repeated independently with similar results.

## Reporting summary

Further information on research design is available in the Nature Portfolio Reporting Summary linked to this article.

# Data availability

Data supporting the findings of this work are available within the paper and its Supplementary Information files. Protein structure files are downloaded from UniProt database (https://www.uniprot.org/), and the identifier of each protein structure we used for protein-protein interaction prediction were listed in Table S2 in the Data Supplement. All primers are listed in the Tables in the Data Supplement. Uncropped images were attached in Supplementary Information. Source data are provided with this paper and uploaded in Figshare (https://doi.org/10.6084/m9.figshare.24204111). Source data are provided with this paper.

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

## Acknowledgements

The authors acknowledge Dr. Bo Liu (Dalian University of Technology) for providing FRET biosensors to visualize cell membrane tension and Dr. Weizhen Zhang (Peking University Health Science Center) for insightful discussions on adipose tissue metabolism assessment. This work was funded by the National Natural Science Foundation of China (#32325030, #82270419, #81921001, and #81974052 to J.Z., #21991133 to X.X., #32201071 to C.Z.), the CAS Strategic Priority Research Program of Chinese Academy of Sciences (XDB37000000 to W.L.), and Shenzhen Bay Scholars Program.

## Author contributions

J.Z., and W.L. supervised the project and conceived the idea. J.L and C.Z. performed the experiments and statistical analysis. X.X. and A.L. provided technical help in magnetic tweezers experiments. Y.L. and J.L. performed plasmid construction. L.F. helped with the cyclic stretch

experiment. J.Z. helped with the flow experiment. Z.L. helped with animal study. W.P. and W.Y. provide discussion on the results. The manuscript was written by J.L. with the guidance of J.Z.

## Competing interests

The authors declare no competing interests.
