## [Peer Review File · Nature Communications]

Endothelial Discoidin Domain Receptor 1 Senses Flow to Modulate YAP ActivationREVIEWER COMMENTS

Reviewer #1 (Remarks to the Author):

In this paper the authors provide evidence that DDR1 is expressed and acts as a mechanosensor in endothelial cells to convey the forces imposed by endothelial shear stress. They propose that shear force causes conformational changes in DDR1 ectodomain that result in receptor oligomerization and phase separation. Overall this is a strong paper and the results presented are novel and important. It is valuable to perform both in vitro and in vivo studies to show changes in receptor activation with atherosclerotic shear stress patterns, and to perform receptor knockout to show the effect on the vascular development of atherosclerosis. In addition the authors have performed extensive and detailed studies of the mechanosensitive molecular signalling induced by DDR1, and valuable and high quality protein-protein interaction studies. It is intriguing that they have identified a mechanism involving interactions with 14-3-3 proteins. However, there are some problems with data quality, and some controls are missing.

Major Points

- The authors have used a VE-cadherin promoter to drive Cre to KO DDR1 in endothelial cells. However, there are reported interactions between cadherins (E and N-cadherin) and DDR1. The authors should determine whether there are interactions between VE cadherin and DDR1 in ECs, and perform controls to determine whether the use of this Cre driver interfere with expression of VE-cadherin/DDR1 interactions to adversely affect endothelial phenotype?
- Although it was confirmed that lipid content in blood was similar in WT and KO cells, is it possible that other cell types and their function could be affected by the Cdh5-specific DDR1 KO? For example, CDH5 is expressed in adipose tissue and DDR1 KO affects the development of adipose tissue in adipose tissue (Lino et al., 2020 Molecular Metabolism). It is well known that changes in adipose are linked to cardiovascular disease outcome. How do the authors know that the decrease in atherosclerosis formation that they see is not attributed to DDR1 metabolic changes in adipose tissue.
- The authors have used super-resolution confocal techniques to view LLPS and DDR1 droplets. However there is little explanation provided as to what these droplets are. How do they relate to lipid rafts, where DDR1 has been shown to localize? What is the relationship to endosomal or exocytotic vesicles? What is the relevance of droplets or LLPS to receptor function?

- There is demonstration of calcium signaling in the ECs when forces are applied by pulling on the DDR1. How do the authors think that DDR1 causes these changes? This deserves discussion.
- Line 192 – these results need a better explanation. This reviewer accepts that the DS-like domain from DDR1 is required for shear dependent activation of DDR1, but the DS domain from DDR1 will suffice for ligand (Col) dependent activation. Why does DDR2 not form droplets? They should include some discussion of why/how DDR2 activation is different from DDR1.
- Figure 2J – thoracic aorta vs aortic arch. In the thoracic aorta, there seems to be more DDR1 and VE-cadherin expression, and it is localized diffusely throughout the cell compared to the aortic arch. In addition, there is more VE-cadherin localized to the cell membrane in the thoracic aorta compared to the aortic arch. Why are these localization patterns different?
- Figure 7 – How does the elimination of both DDR1 (by inhibitor treatment) and 14-3-3 ϵ (YWHAE by siRNA) reverse phenotype to wildtype (ie YAP localizes to nucleus and stimulates target gene expression)? DDR1 inhibition prevents YAP translocation to the nucleus (likely due to 14-3-3 ϵ triggering YAP degradation or preventing its translocation through binding) but removal of 14-3-3 ϵ at the same time allows YAP to return to the nucleus. What is the upstream stimulus for YAP activation in this context? Does DDR1 influence translocation but is not necessary/sufficient? What happens if you knockdown 14-3-3 ϵ only and allow DDR1 expression? Do you see the same results? Does DDR1 interact with and influence LATS1/2 in endothelial cells?

Minor points

- There is little detail provided on methods and conditions used to characterize different patterns of shear. How is oscillatory shear accomplished in a linear microfluidic chamber? It is not clear how cell orientation was measured, nor is it clear what the coloured ray plots indicate in figure 1.
- Figure 7D – it looks like cells treated with OS and DDR1in1 still have DDR1 droplets. Why? Were all the experiments in figure 7 done in HUVECs? The intensity and pattern of staining differs between 7A and 7D and E.
- Figure 4G and I – the time zero image in Figure 4G look very blurred compared to 15 min and 30 min images. It looks like the picture just gets clearer as time goes on without a change in intensity for aggregates. Figure 4I in the C287A mutated group, images are not very clear, very fuzzy/blurry. The authors should provide better images.
- Figure 5 and 6 – Co-IPs Figure 5B and Figure 6D both include molecular weight markers with the western blots, whereas all other western blots do not. Please provide molecular weight markers on all

blots. If the journal requires, it would be useful to provide all uncropped western blots, in particular because DDR1 is cleaved giving rise to lower MW species.

Reviewer #2 (Remarks to the Author):

The manuscript by Liu et al described the role of discoidin domain receptor 1 (DDR1) tyrosine kinase as a direct mechanosensor and essential in conveying the force imposed by shear to the endothelial responses. A very nice study with comprehensive set of experiments.

I have the following comments on the paper:

- Why, HUVECs? What is the range of shear stress that vein endothelial cells experience? Do they experience oscillatory flow? Why not aortic endothelial cells? I recommend comparing the response of HAECs to the vein cell lines for figure 1A and B potentially other in vitro studies or at least the key experiments.

- Does shear stress affects the expression of DDR1 in endothelial cells?

Shear induced droplet formation of DDR1. What does it mean? Does it correlate with the activity of DDR1?

- Figure 2G: the shear-induced DDR1 droplets formed on the apical side of the cells, rather than on the basal side (Figure 2G);

How can this figure dissociate between apical membrane and basal membrane? For the basal membrane you will need to use the live imaging approach and TIRF microscopy. For apical membrane, probably you will need live imaging and upright microscopy focussing on the luminal surface. From this figure, you can not draw this conclusion.

- Figure 2I-J: DDR1 form a complex with VE-cadherin. Can you assess the effect of flow on the complexation of DDR1 with VE-cadherin in your in-vitro model. Does the complexes of DDR1 contains VE-cadherin as well? Ex. In Fig 2G-H.

- Does ECM proteins affect the DDR1 complexation/ activation? What ecm protein have been used to cover the channels for the in vitro experiments?

- Line 142. To assess and identify the contribution of Piezo1, I recommend use siRNA knock down approach, to knockdown piezo1 in endothelial cells and repeat figure 3B.

Since you are detecting increase in $[Ca^{2+}]$, there also a possibility that DDR1 is activated downstream of MS Ca^{2+} channel such as Piezo1.

- This paper also needs a summary figure proposing how DDR1 is regulating mechanotransduction of shear.

- Lit review is not complete. Earlier studies on endothelial mechanotransduction should also be included in the introduction and discussion including the followings: DOI: 10.1111/brv.12814, DOI: 10.1111/brv.12814, DOI: 10.1111/brv.12814

Dear Reviewers,

Thank you very much for your time involved in reviewing the manuscript and your very encouraging comments on the merits.

Reply to Reviewer #1:

• The authors have used a VE-cadherin promoter to drive Cre to KO DDR1 in endothelial cells. However, there are reported interactions between cadherins (E and N-cadherin) and DDR1. The authors should determine whether there are interactions between VE cadherin and DDR1 in ECs, and perform controls to determine whether the use of this Cre driver interfere with expression of VE-cadherin/DDR1 interactions to adversely affect endothelial phenotype?

Response: We express our appreciation for the invaluable insights provided by the reviewers. The Cdh5(PAC)-CreER^{T2} mouse model employed in our study was generated by introducing a transgene containing a genomic Cdh5(PAC) promoter fragment fused to a CreER^{T2} cDNA into C57BL/6 zygotes. This innovation approach ensures independent expression of both Cre enzyme and VE-cadherin (Cdh5), distinct from directly inserting CreER^{T2} into the Stop codon of Cdh5. Thus, the utilization of this Cre driver does not affect the expression of VE-cadherin.

Due to the limited yield achievable from isolating and purifying ECs from mice, the execution of immunoprecipitation experiments presents significant challenges. Consequently, we chose an alternative route, focusing on investigating potential interactions between VE-cadherin and DDR1 through techniques such as immunofluorescence staining and co-immunoprecipitation (Co-IP) assays conducted using HUVECs. Notably, live cell imaging revealed distinct distribution patterns for DDR1 and VE-cadherin on the cell membrane under static conditions. Interestingly, when DDR1 was activated, the co-localization of DDR1 and VE-cadherin was scarcely observed in both live and fixed cells (Supplementary Figure S6A-D). The findings from Co-IP experiments further confirmed the absence of interaction between DDR1 and VE-cadherin in HUVECs (Supplementary Figure S6E).

Supplementary Figure 6. DDR1 has no interaction with VE-cadherin in endothelial cells.

(A) Live cell imaging of DDR1-EGFP and VE-cadherin-Cherry in ECs, which were grown in microfluidic chamber and subjected to laminar shear stress. (B) Colocalization analysis of DDR1-EGFP and VE-cadherin-Cherry in (A). Pearson's R value (above threshold) was calculated by ImageJ Fiji software (Analyze-colocalization-coloc2). $n = 12$ images from 3 biological replicates. (C) Immunofluorescence staining of DDR1 and VE-cadherin in HUVECs subjected to laminar shear stress for 1h or static. (D) Colocalization analysis of DDR1 and VE-cadherin in (C). $n = 12$ images from 3 biological replicates. Data were all expressed as the means \pm SEM and analyzed by Mann-Whitney test. (E) Co-IP assay to detect the interaction between DDR1 and VE-cadherin in HUVECs subjected to PS or OS for 24h. Scale bars, 20 μ m.

• Although it was confirmed that lipid content in blood was similar in WT and KO cells, is it possible that other cell types and their function could be affected by the *Cdh5*-specific DDR1 KO? For example, CDH5 is expressed in adipose tissue and DDR1 KO affects the development of adipose tissue in adipose tissue (Lino et al., 2020 *Molecular Metabolism*). It is well known that changes in adipose are linked to cardiovascular disease outcome. How do the authors know that the decrease in atherosclerosis formation that they see is not attributed to DDR1 metabolic changes in adipose tissue.

Response: We greatly appreciate the insightful suggestion provided by the reviewer. Notably, prior research has indicated that a systemic knockout of DDR1 can lead to a reduction in obesity and an increase in brown fat activity (Lino et al., 2020 *Molecular Metabolism*). Given this context, it becomes crucial to determine whether targeted knockout of DDR1 specifically in *Cdh5*-positive cells (ECs) would impact lipid metabolism. In our pursuit of understanding the potential implications of *Cdh5*-specific DDR1 knockout on adipose tissue metabolism, we initially checked the expression of *Cdh5* across different cell types within adipose tissue using Tabula Muris. This resource compiles single-cell transcriptome data from the model organism *Mus musculus*. The data derived from single-cell sequencing of adipose tissue revealed a predominant expression of *Cdh5* in endothelial cells.

Single-cell sequencing of adipose tissue reveals a predominant expression of *Cdh5* in endothelial cells.

(A and B) t-SNE plot showed *Cdh5* expression in different cell subsets of fat tissue (A) and cell annotations in seven clusters (B).

To delve further into the potential effects of DDR1 knockout in ECs on adipose tissue metabolism, we collected the epididymal white adipose tissue (eWAT) and subcutaneous white adipose tissue (scWAT) from *Ddr1*^{WT} and *Ddr1*^{iECKO} mice that were fed a Western diet. Subsequently, we conducted qPCR analysis to assess the expression of genes involved in lipid synthesis (*Dgat1*, *Gpam*, *Scad1*, and *Pparg*),

fatty acid transport (*Cd36*, *Fatp1*, *Fabp1*, and *Fatp4*), and lipolysis (*Macd*, *Lcad*, and *Vlcad*). Additionally, we investigated the thermogenic activity of brown adipose tissue (BAT) by examining the expression of *Ucp1*, *Prdm16* and *Pgc1a*. The findings, as illustrated in the accompanying figures (Supplementary Figure S3), along with the data demonstrating comparable lipid content in the blood of both WT and KO mice, indicate that *Cdh5*-specific DDR1 knockout does not exert a discernible influence on lipid metabolism.

Supplementary Figure 3. *Cdh5*-specific DDR1 knockout does not exert a discernible influence on lipid metabolism.

(A) The weight-to-body weight ratio of epididymal white adipose tissue (eWAT), subcutaneous white adipose tissue (scWAT) and brown adipose tissue (BAT) in western-diet-fed *Ddr1*^{WT} and *Ddr1*^{iECKO} mice. (B-D) RT-qPCR to detect lipid synthesis (B), fatty acid transport (C) and lipolysis (D) of eWAT. n=4 mice. (E-G) RT-qPCR to detect lipid synthesis (E), fatty acid transport (F) and lipolysis (G) of scWAT. n=4 mice. (H) RT-qPCR to detect the thermogenesis of BAT. n=4 mice. Data were all expressed as the means ± SEM and analyzed by Mann-Whitney test.

•The authors have used super-resolution confocal techniques to view LLPS and DDR1 droplets. However there is little explanation provided as to what these droplets are. How do they relate to lipid rafts, where DDR1 has been shown to localize? What is the relationship to endosomal or exocytotic vesicles? What is the relevance of droplets or LLPS to receptor function?

Response: We appreciate the reviewer's insightful observations. The observed DDR1 droplets are membraneless organelles that arise from the activation of DDR1. These droplets originate from the activated DDR1 molecules initially present at the cell membrane. We have added more descriptions of the DDR1 droplets into the revised manuscript.

In relation to the localization of DDR1 with lipid rafts, a key point of interest, we conducted immunostaining experiments involving DDR1 and lipid raft marker caveolin-1 (Cav-1). Our findings revealed that the majority of DDR1 under static conditions is situated within the lipid raft regions. However, as a response to shear stress, DDR1 gradually disengages from the lipid raft and forms droplets (Supplementary Figure S5A). This observation provides valuable insights into the dynamic interplay between DDR1 and lipid raft structures.

Supplementary Figure 5. (A) Immunofluorescence of DDR1-EGFP and lipid raft marker caveolin-1 (Cav-1)

in HUVECs, which were infected with DDR1-EGFP adenovirus and subjected to laminar shear. Scale bars, 10 μm .

Regarding the potential relationship between DDR1 droplets and endosomal or exocytotic vesicles, it's important to highlight the distinction between membranous organelles like endosomes and extracellular vesicles and membraneless structures such as condensate droplets. Although a notable distinction exists between phase-separated droplets and vesicles due to their membrane characteristics, it has been documented that droplets can indeed become incorporated into endosomes (*J Cell Biol.* 2022 Oct;221(10):e202203102). Our previous studies have also demonstrated that DDR1 droplets can undergo endocytosis in vascular smooth muscle cells (*Bioact Mater.* 2022 Nov; 17: 406 - 424. *Circ Res.* 2023 Jan;132(1):87-105). To explore this further, we conducted immunofluorescence experiments to detect DDR1 and the early endosome marker Rab5A in ECs. The results indicated that only a limited number of DDR1 droplets colocalized with Rab5A (Supplementary Figure S5B), suggesting a potential but not extensive interaction with early endosome. Moreover, considering the critical role of late endosome-lysosome fusion in the degradation of endocytosed cargo, we employed Lyso-Tracker Red to label lysosomes. Intriguingly, our observations reveal that the majority of DDR1 droplets do not exhibit colocalization with lysosomes (Supplementary Figure S5C). These findings collectively provide insights into the intricate dynamics between DDR1 droplets and various intracellular compartments.

Supplementary Figure 5. (B) Immunofluorescence of DDR1-EGFP and early endosome marker Rab5A in HUVECs, which were infected with DDR1-EGFP adenovirus and subjected to laminar shear for 1h. (C) Immunofluorescence of DDR1-EGFP and Lyso-Tracker Red in HUVECs, which subjected to laminar shear for 1h. Scale bars, 10 μm .

To date, there is no documented research indicating the incorporation of droplets into extracellular vesicles. However, a recent cryoET survey investigating the molecular architecture of a β -amyloid plaque revealed the presence of extracellular droplets within these plaques (*Nat Commun.* 2023 May;14(1):2833). Furthermore, previous studies have reported the detection of DDR1 within

exosomes (*J Egypt Natl Canc Inst.* 2015 Jun;27(2):51-58). To delve into the potential relationship between DDR1 and exocytotic vesicles, we conducted immunofluorescence experiments involving DDR1 and exosomes marker CD63. The results revealed that a minority of DDR1 droplets could indeed be incorporated into extracellular vesicles (Supplementary Figure S5D). Drawing from this evidence, while some DDR1 droplets exhibit the ability to integrate into endosomal or exocytotic vesicles, the majority of DDR1 droplets persist in the state of membraneless organelles.

Supplementary Figure 5. (D) Immunofluorescence of DDR1-EGFP and exosomes marker CD63 in HUVECs, which subjected to laminar shear for 1h. Scale bars, 10 μm .

As for the significance of droplets or liquid-liquid phase separation (LLPS) in relation to receptor function, the efficient recruitment of signaling proteins to activated receptor tyrosine kinases (RTKs) to generate rapid and precise downstream responses is often compromised by the stochastic nature of random diffusion to the target site. The concept of LLPS addresses this challenge by facilitating localized and heightened concentrations of the necessary proteins, concurrently hindering competing ligands (*Mol Cell.* 2022 Mar 17;82(6):1089-1106.e12).

•There is demonstration of calcium signaling in the ECs when forces are applied by pulling on the DDR1. How do the authors think that DDR1 causes these changes? This deserves discussion.

Response: Thanks for the insightful suggestion from the reviewer. A prior study highlighted the ability of tyrosine kinase receptors (TKRs) activation to trigger calcium influx in bovine aortic endothelial cells (*J Cell Physiol.* 2000 Dec;185(3):454-463). The study established that the heightened calcium signaling resulting from TKRs activation originated exclusively from extracellular Ca^{2+} entry. Furthermore, they demonstrated that two factors (bFGF and IGF-I) which interact TKRs, could activate calcium-permeable channels. Additionally, literature has indicated that the phosphorylation of TKRs on tyrosine residues serves as a crucial mechanism for governing calcium entry (*FEBS Lett.* 1995 Aug 14;370(1-2):127-30 & *J Physiol.* 1995 May;484 (Pt 3):557-566). Drawing upon the insights from the references mentioned above and our own discoveries, we postulated that DDR1 activation induces calcium influx through receptor phosphorylation, subsequently activating calcium-permeable channels. The above discussion has been incorporated into the revised manuscript (Page 11, Lines 3~9).

- **Line 192** - these results need a better explanation. This reviewer accepts that the DS-like domain from DDR1 is required for shear dependent activation of DDR1, but the DS domain from DDR1 will suffice for ligand (Col) dependent activation. Why does DDR2 not form droplets? They should include some discussion of why/how DDR2 activation is do different from DDR1.

Response: We greatly appreciate the insightful feedback provided by the reviewer. According to our current findings, the DS-like domain of DDR1 assumes the role of force perception, while the prior investigations have demonstrated that the DS domain of DDR1 is pivotal for collagen binding (*Int Rev Cell Mol Biol.* 2014;310:39-87). Despite sharing a certain degree of sequence identity in the DS (59% identity) and DS-like (51% identity) domains, DDR1 and DDR2 exhibit notable structural differences, particularly in the DS-like domain. The accompanying figure portrays the crystal structure of the extracellular domains of DDR1 and DDR2, highlighting disparities in the DS-like domain through red dashed lines. This disparity could potentially account for the unresponsiveness of DDR2 to shear stress.

Crystal structure of the extracellular domains of DDR1 and DDR2.

Regarding collagen-induced droplets formation, it has been documented that DDR2-GFP fails to undergo receptor clustering upon collagen stimulation, manifesting solely as filamentous structures rather than punctuate formations (*J Mol Biol.* 2019 Jan 18;431(2):368-390). Our earlier investigation indicated that the transmembrane (TM) domain of DDR1 governs the assembly of globular clusters and LLPS (*Circ Res.* 2023 Jan;132(1):87-105). Discrepancies in the domain that substantially contributes to protein phase separation could conceivably underlie the differences observed in the self-assembly patterns of DDR1 and DDR2. These deliberations have been thoughtfully integrated into the revised manuscript (Page 6, Lines 30~32 and Page 7, Lines 3~4).

- **Figure 2J** - thoracic aorta vs aortic arch. In the thoracic aorta, there seems to be more

DDR1 and VE-cadherin expression, and it is localized diffusely throughout the cell compared to the aortic arch. In addition, there is more VE-cadherin localized to the cell membrane in the thoracic aorta compared to the aortic arch. Why are these localization patterns different?

Response: We appreciate the insightful observations made by the reviewer. The contrasting localization patterns of DDR1 and VE-cadherin in the thoracic aorta and aortic arch can indeed be attributed to specific flow conditions. Numerous studies have established that disturbed flow is capable of triggering endothelial-to-mesenchymal transition, characterized by a reduced expression of VE-cadherin (*Cell Rep.* 2020 Dec;33(11):108491). Moreover, oscillatory shear simulating the disturbed flow has been associated with the downregulation of VE-cadherin expression (*Cell Mol Biol Lett.* 2023 Mar;28(1):22). It is worth noting that disturbed flow has the potential to induce significant changes, including the rearrangement and internalization of VE-cadherin (*Front Cell Dev Biol.* 2022 Nov 10;10:963866), which in turn can disrupt the VE-cadherin/catenin complex (*Eur Heart J.* 2023 Jan 21;44(4):304-318). Thus, the divergence in flow patterns is a likely determinant of the distinct localization patterns observed for VE-cadherin.

• Figure 7 - How does the elimination of both DDR1(by inhibitor treatment) and 14-3-3 ϵ (YWHAE by siRNA) reverse phenotype to wildtype (ie YAP localizes to nucleus and stimulates target gene expression)? DDR1 inhibition prevents YAP translocation to the nucleus (likely due to 14-3-3 ϵ triggering YAP degradation or preventing its translocation through binding) but removal of 14-3-3 ϵ at the same time allows YAP to return to the nucleus. What is the upstream stimulus for YAP activation in this context? Does DDR1 influence translocation but is not necessary/sufficient? What happens if you knockdown 14-3-3 ϵ only and allow DDR1 expression? Do you see the same results? Does DDR1 interact with and influence LATS1/2 in endothelial cells?

Response: We appreciate the thoughtful insights provided by the reviewer regarding Figure 7 and its implications. In the context of inhibiting DDR1 alone, our findings clearly indicated that the YAP nuclear translocation triggered by oscillatory shear (OS) was significantly attenuated. While we cannot assert that DDR1 is the sole necessary or sufficient receptor for OS-induced YAP activation, we can confidently affirm that DDR1 plays a pivotal role as an upstream mediator of YAP signaling.

Upon simultaneous inhibition or elimination of both DDR1 and 14-3-3 ϵ (YWHAE), we observed the reactivation of YAP induced by OS. This outcome can be attributed to 14-3-3 ϵ elimination that preserves YAP in the cytoplasm by preventing its degradation. The dissociation of 14-3-3 proteins from phosphorylated YAP permits YAP dephosphorylation and subsequent nuclear translocation (*Int Immunopharmacol.* 2023 Jun;119:110220). To provide further insights, we have incorporated

immunostaining to visualize the intracellular distribution of YAP in the case of exclusively knocking down 14-3-3 ϵ (Supplementary Figure S12A-B). Our results demonstrate that the knockdown of 14-3-3 ϵ significantly enhanced YAP nuclear translocation in the presence of PS conditions. However, in the context of OS, the impact of 14-3-3 ϵ knockdown on YAP seems less significant. These discrepancies could potentially be attributed to the role of 14-3-3 ϵ suppression in mediating the OS-induced YAP signaling.

Supplementary Figure 12. The elimination of YWHAE promotes the nuclear translocation of YAP.

(A) Representative Immunofluorescence staining of DDR1 and YAP1 in HUVECs subjected to PS or OS for 24 hours. (B) Quantification of YAP localization in (A). $n=9$ images from 3 biological replicates. 10~15 cells per image. Data were expressed as the means \pm SEM and analyzed by two-way ANOVA followed by Tukey's multiply test. Scale bar, 20 μ m.

As for the interaction between DDR1 and LATS1, we have observed their colocalization in HUVECs subjected to PS or OS (Supplementary Figure S13A-B). Colocalization analysis has indicated variations in the colocalization factor of DDR1 and LATS1 over different time periods of shear stress. In an effort to elucidate the role of the DDR1-LATS1 axis in mediating OS-induced YAP activation, we conducted experiments using a combination of DDR1 inhibitor and LATS inhibitor. Immunostaining results revealed that LATS inhibition using Lats-IN-1 could partially mitigate the inhibitory effect of DDR1-IN-1 on YAP activation (Supplementary Figure S13C-D). This insight underscores the partial involvement of the DDR1-LATS1 interaction in the regulatory process of YAP

by DDR1. It is important to note, however, that our focus on the DDR1-LATS1 interaction has been relatively limited, given that LATS1 did not emerge in our mass spectrometry results.

Supplementary Figure 13. DDR1 can interact with LATS1 to regulate YAP nuclear translocation partially.

(A) Representative Immunofluorescence staining of DDR1 and LATS1 in HUVECs subjected to PS or OS for four time periods (15 min, 30 min, 1 h and 3 h). (B) Co-location analysis of DDR1 and LATS1. Manders' colocalization coefficients was calculated by ImageJ Fiji software (Analyze-colocalization-coloc2), representing the proportion of LATS1 (co-located with DDR1) to the total LATS1. n=6 images from 3 biological replicates. Data were expressed as the means \pm SEM and analyzed by Kruskal-Wallis test with Dunn's test. (C) Immunofluorescence and quantification of YAP localization in HUVECs. The cells were subjected to PS or OS

for 24 hours, and treated with inhibitors or DMSO. $n = 9$ images from 3 biological replicates. 10~15 cells per image. Data were expressed as the means \pm SEM and analyzed by two-way ANOVA followed by Tukey's multiply test. Scale bars, 20 μm .

Minor points

- **There is little detail provided on methods and conditions used to characterize different patterns of shear. How is oscillatory shear accomplished in a linear microfluidic chamber?**

Response: We appreciate the reviewer's insightful input regarding the methods and conditions employed for characterizing various shear patterns. Within our microfluidic chamber, the application of geometric obstruction leads to a remarkable ≈ 100 -fold amplification in shear stress at the stenotic site. This results in the establishment of distinct shear stress patterns. Specifically, the region characterized by elevated and unidirectional shear stress, situated at the stenotic area, is deemed to be representative of a laminar flow environment. Conversely, just downstream of the narrowed region, a region with lower and reciprocating shear stress prevails, indicative of a disturbed flow area. For the execution of oscillatory shear within a linear microfluidic chamber, computational fluid dynamics simulations have been instrumental in portraying the shear rate distribution. To provide visual clarity, we have included a figure showcasing these simulations, where the disturbed flow region is highlighted using a red dashed line (*Science*. 2012 Aug 10;337(6095):738-42). This figure substantiates our characterization of the distinct shear patterns observed in our microfluidic setup. These details have been thoughtfully integrated into the Expanded method (Page 2, lines 7~12), enriching the description of the methods and conditions utilized in our study. We sincerely thank the reviewer for the opportunity to enhance the clarity of our explanation.

Hemodynamic conditions in microfluidic devices.

(A) A photograph of the microdevice that mimics vascular stenosis fabricated in PDMS. (B) Computational fluid dynamics (CFD) simulations of the microfluidic device demonstrating that geometric obstruction generates a ≈ 100 -fold increase in shear stress at the stenotic site. The just downstream of the narrowed region with low and reciprocating shear stress is a disturbed flow area.

- **It is not clear how cell orientation was measured, nor is it clear what the coloured ray plots indicate in figure 1.**

Response: Thanks for the insightful suggestion from the reviewer regarding the measurement of cell orientation and the interpretation of the colored ray plots in Figure 1. To analyze cell orientation, we employed the OrientationJ plugin within the Fiji software to determine the orientation of actin filaments. In a nutshell, the image data was initially subjected to skeletonization using a Hessian-based multiscale filter, followed by a conversion into a binary label matrix. Subsequently, structural elements were fitted to the skeleton, and segments smaller than 5 pixels were disregarded. The maximum Feret diameter of each segmented filament was identified as the longest possible distance between two parallel tangents at opposing structural element borders. The angle between the maximum Feret diameter relative to the direction of applied flow was then calculated. The colored ray plots in Figure 1 serve to illustrate the cumulative distribution of filaments within specific angle ranges relative to the flow direction, categorized as 0-10, 11-20, 21-30, 31-40, 41-50, 51-60, 61-70, 71-80 or 81-90, respectively. To provide a comprehensive account of these methodologies, we have included an explanation within the Expanded Methods section (Page 1, Lines 26~36).

- **Figure 7D - it looks like cells treated with OS and DDR1-IN-1 still have DDR1 droplets. Why?**

Response: Thanks for the insightful suggestion from the reviewer regarding Figure 7D. Indeed, it is evident that cells treated with oscillatory shear (OS) and DDR1-IN-1 still exhibit DDR1 droplets. The underlying mechanism for this observation can be attributed to the nature of DDR1-IN-1 and the factors governing DDR1 droplet formation. Previous research has documented that DDR1-IN-1 can bind to the intracellular kinase domain of DDR1, leading to the inhibition of DDR1 autophosphorylation (*ACS Chem Biol.* 2013 Oct 18;8(10):2145-50). While it has been established that the phosphorylation status of TRKs could influence the LLPS of TKRs, as seen in cases such as EGFR and FGFR (*Proc Natl Acad Sci U S A.* 2022 May 10;119(19):e2122531119. *Mol Cell.* 2022 Mar 17;82(6):1089-1106.e12), our own previous research identified that the transmembrane domain is a crucial determinant for DDR1 droplet formation (*Circ Res.* 2023 Jan;132(1):87-105). Consequently, while DDR1-IN-1 treatment may indeed impact some aspects of DDR1 function, it cannot fully abrogate the process of DDR1 droplet formation due to the intricate interplay of factors driving this phenomenon.

- **Were all the experiments in figure 7 done in HUVECs? The intensity and pattern of staining**

differs between 7A and 7D and E.

Response: We appreciate the reviewer's insightful observation regarding the experiments in Figure 7. All experiments depicted in Figure 7 were indeed conducted using HUVECs. The variations in staining intensity and patterns between Figures 7A and 7D-E can be attributed to differences in magnifications and cellular heterogeneity. The variations in cellular morphology and distribution might result in differences in the apparent staining intensity and arrangement. Additionally, even though antibodies share the same serial number, variations in their lot number can also influence the staining outcome. To address these concerns, we have taken steps to enhance clarity. We replaced Figure 7A with an image taken at a magnification that better captures a broader view of cells. Furthermore, we ensured that the immunostaining for Figure 7D and 7E was performed using the same antibody, mitigating any potential variations introduced by different antibodies.

Figure 7. Immunofluorescence of YAP localization in HUVECs. The cells were subjected to PS or OS for 24 hours. Scale bars, 20 μm .

• **Figure 4G and I** – the time zero image in Figure 4G look very blurred compared to 15 min and 30 min images. It looks like the picture just gets clearer as time goes on without a change in intensity for aggregates. Figure 4I in the C287A mutated group, images are not very clear, very fuzzy/blurry. The authors should provide better images.

Response: Thanks for the valuable suggestion from the reviewer. We apologized for not carefully checking the image clarity. Accordingly, we have replaced Figure 4G and 4I with clearer images.

Figure 4. (G) Live cell imaging of DDR1 WT or locked mutant in ECs after 10µg/ml soluble collagen I stimulation, and quantification of DDR1 condensates number. (I) Live cell imaging of DDR1 WT or C287A mutant in ECs after 10µg/ml soluble collagen I stimulation, and quantification of DDR1 condensates number. Scale bars, 10 µm.

• **Figure 5 and 6 - Co-IPs Figure 5B and Figure 6D both include molecular weight markers with the western blots, whereas all other western blots do not. Please provide molecular weight markers on all blots. If the journal requires, it would be useful to provide all uncropped western blots, in particular because DDR1 is cleaved giving rise to lower MW species.**

Response: Thanks for the valuable suggestion from the reviewer. We have included molecular weight markers within all the non-reducing Western blots (Figure 2C and 4L) to better demonstrate the different aggregation states of DDR1. Taking Figure S1B as an example, all presented immunoblots of DDR1 had a molecular weight near 140 kD in the reducing SDS-gel electrophoresis, demonstrating the detected DDR1 was full length. Considering the limited space, we only denoted the location of molecular weight standards in other immunoblots and provided all uncropped blots in the supplementary file-uncropped gel.

Figure S1B. (B) The knockdown efficiency of DDR1 confirmed by western blotting in HUVECs transfected with siRNAs specific for DDR1 or scrambled siRNA. Two biological replicates were showed.

Reply to Reviewer #2:

• **Why, HUVECs? What is the range of shear stress that vein endothelial cells experience? Do they experience oscillatory flow? Why not aortic endothelial cells? I recommend comparing the response of HAECs to the vein cell lines for figure 1A and B potentially other in vitro studies or at least the key experiments.**

Response: We appreciate the reviewer's insightful query regarding the use of HUVECs (human umbilical vein endothelial cells) in our study. The choice of HUVECs stems from their physiological relevance in carrying oxygenated blood/arterial blood from the placenta to the growing fetus through the relatively straight and elongated umbilical vein. In healthy cases, the blood flow velocity in umbilical vein remains steady, approximately 5.2 dyne/cm^2 (*Cell Biol Int.* 2021 Sep;45(9):1926-1934). Consequently, endothelial cells in umbilical vein don't experience oscillatory shear stress. However, the rationale for utilizing HUVECs in our study is grounded in previous investigations (*Proc Natl Acad Sci U S A.* 2021 Jan 26;118(4):e2023236118 & *Proc Natl Acad Sci U S A.* 2012 May 15;109(20):7770-5). These studies have demonstrated that, when examining pathways to pulsatile or laminar shear versus oscillatory shear in both human aortic ECs (HAECs) and HUVECs, the outcomes are qualitatively similar with a few exceptions. These findings have substantiated that HUVECs can serve as a suitable model for investigating the effects of shear stress on arterial endothelial cells, thus justifying their use in our experimental setup.

• **Does shear stress affects the expression of DDR1 in endothelial cells? Shear induced droplet formation of DDR1. What does it mean? Does it correlate with the activity of DDR1?**

Response: We appreciate the reviewer's insightful queries regarding the effects of shear stress on DDR1 expression and the implications of DDR1 droplet formation. Allow us to address these points more clearly.

In response to the reviewer's question regarding the influence of shear stress on the expression of DDR1 in endothelial cells (ECs), we have conducted Western blotting and qPCR experiments to assess the expression of DDR1 in ECs exposed to PS or OS for 24h. The results indicated no discernible difference in DDR1 expression between the two groups (Supplementary Figure S4A-B). This suggests that shear stress, as experienced in the specific conditions tested, did not significantly influence the expression of DDR1 in these ECs.

The concept of DDR1 droplets formation pertains to the phenomenon of liquid-liquid phase separation (LLPS) of DDR1. Upon DDR1 activation, multiple tyrosyl phosphate (pY) binding sites with moderate affinity become accessible for the rapid recruitment of downstream effector proteins. LLPS, in this

context, enables the creation of localized, elevated concentrations of necessary proteins while impeding competing ligands. This process has been established to function as a regulatory switch for the enzyme activity of receptor tyrosine kinases (*Nature*. 2012 Mar 7;483(7389):336-40). Within the condensed state characterized by apparent local concentration increases, the highly charged, multivalent environment of a droplet potentially facilitates the attainment of catalytically favorable conformations (*Mol Cell*. 2022 Mar 17;82(6):1089-1106.e12). Consequently, the formation of DDR1 droplets is intrinsically linked to DDR1 kinase activity and downstream functional effects.

Figure S4. Shear stress does not affect the expression of DDR1 in endothelial cells.

(A) Western blotting to detect DDR1 expression in HUVECs subjected to PS or OS for 24h. 3 biological replicates were presented. (B) Quantitative RT-PCR analysis of the expression of DDR1 in HUVECs treated as in (A). n=4 biological replicates. Data were expressed as the means \pm SEM and analyzed by Mann-Whitney test.

• **Figure 2G: the shear-induced DDR1 droplets formed on the apical side of the cells, rather than on the basal side (Figure 2G);**

How can this figure dissociate between apical membrane and basal membrane? For the basal membrane you will need to use the live imaging approach and TIRF microscopy. For apical membrane, probably you will need live imaging and upright microscopy focussing on the luminal surface. From this figure, you can not draw this conclusion.

Response: We appreciate the reviewer's insightful observation regarding Figure 2G and the distinction between apical and basal sides of the cells. It is indeed crucial to accurately differentiate between these cellular surfaces in order to draw meaningful conclusions. Our approach in this experiment involved seeding HUVECs on the bottom surface of microfluidics chamber, followed by vertical cross-sectional imaging. To provide a robust assessment of the cellular surfaces to distinguish between the apical and basal membranes, we have undertaken live imaging with 3D reconstruction to further elucidate the localization of DDR1 droplets (Supplementary video S1 and S2). By focusing on the curvature of the cell and the endothelial cell morphology, the apical and basal surfaces can be discerned (*Fluids Barriers CNS*. 2023 Jan 9;20(1):2). Specifically, the surface in direct contact with the flat bottom of

the chamber corresponds to the basal membrane, while the curved surface in contact with the medium and directly exposed to shear stress represents the apical side.

- **Figure 2I-J: DDR1 form a complex with VE-cadherin. Can you assess the effect of flow on the complexation of DDR1 with VE-cadherin in your in-vitro model. Does the complexes of DDR1 contains VE-cadherin as well? Ex. In Fig 2G-H.**

Response: Thanks for the valuable suggestion from the reviewer regarding the interaction of DDR1 with VE-cadherin and the potential influence of flow on this complexation. In our study, we assessed the colocalization of DDR1 and VE-cadherin using Pearson's correlation coefficient in Figure 2K. The analysis indicated minimal colocalization between DDR1 and VE-cadherin. Furthermore, we explored the interaction between DDR1 and VE-Cadherin through live cell imaging and immunofluorescence staining (Supplementary Figure S6A-B). These experiments revealed that while both DDR1 and VE-cadherin were present on the cell membrane in the static conditions, their distribution patterns diverge. Critically, upon DDR1 activation, the colocalization of DDR1 and VE-cadherin was scarcely observed in both live cell and fixed cell contexts. These findings highlight the distinction between the spatial arrangement of DDR1 and VE-cadherin and the limited colocalization observed under the influence of DDR1 activation.

Figure S6. DDR1 has no interaction with VE-cadherin in endothelial cells.

(A) Live cell imaging of DDR1-EGFP and VE-cadherin-Cherry in ECs, which were grown in microfluidic chamber and subjected to laminar shear stress. (B) Colocalization analysis of DDR1-EGFP and VE-cadherin-Cherry in (A). Pearson's R value (above threshold) was calculated by ImageJ Fiji software (Analyze-colocalization-coloc2). n = 12 images from 3 biological replicates. (C) Immunofluorescence staining of DDR1 and VE-cadherin in HUVECs subjected to laminar shear stress for 1h or static. (D) Colocalization analysis of DDR1 and VE-cadherin in (C). n = 12 images from 3 biological replicates. Data were all expressed as the means \pm SEM and analyzed by Mann-Whitney test.

• Does ECM proteins affect the DDR1 complexation/ activation? What ecm protein have been used to cover the channels for the in vitro experiments?

Response: Thanks for the valuable comment from the reviewer concerning the impact of ECM proteins on DDR1 complexation and activation. It is indeed well-established that ECM proteins, particularly collagen, can influence the complexation and activation of DDR1 (*Biochim Biophys Acta*. 2013 Oct;1834(10):2187-94). Furthermore, recognizing that fibronectin has the potential to induce an atherogenic inflammation phenotype in ECs (*Arterioscler Thromb Vasc Biol*. 2018 Nov;38(11):2601-2614), we opted to employ relatively inert gelatin for covering the channels, dishes, and chambers in our vitro experiments. This choice aimed to avoid the potentially confounding influence of collagen or fibronectin on our experimental outcomes.

• Line 142. To assess and identify the contribution of Piezo1, I recommend use siRNA knock down approach, to knockdown piezo1 in endothelial cells and repeat figure 3B. Since you are detecting increase in [Ca²⁺], there also a possibility that DDR1 is activated downstream of MS ca²⁺ channel such as Piezo1.

Response: We greatly appreciate the reviewer's insightful recommendation pertaining to assessing the contribution of Piezo1 and its potential relationship with DDR1 activation. In response to the suggestion, we conducted a knockdown experiment targeting Piezo1 in endothelial cells, followed by the replication of Figure 3B and 3C. This iteration of the experiments demonstrated that the force-induced formation of DDR1 droplets was remains unaffected by Piezo1 knockdown (Supplementary Figure S9A-C). Moreover, we extended our investigation by employing the microfluidic model, which enabled us to confirm that shear stress-induced DDR1 droplet formation is also independent of Piezo1 (Supplementary Figure S9D-E). We sincerely value the reviewer's input, which has led us to provide a more comprehensive analysis of the potential connection between Piezo1 and DDR1 activation.

Figure S9. Knocking down piezo1 does not affect the activation of DDR1.

(A) The knockdown efficiency of DDR1 in HUVECs assessed by RT-qPCR. $n=4$ biological replicates. Data were expressed as the means \pm SEM and analyzed by Mann-Whitney test. (B and C) Images of DDR1 condensates formation captured by total internal reflection fluorescence (TIRF) microscopy. HUVECs were transfected with Piezo1-specific siRNA and infected with DDR1 adenovirus. After 48 hours, cells were incubated with collagen/ BSA-coated beads (B) or anti-DDR1/IgG-coated beads (C). Scale bar, 5 μ m. (D) Time-lapse images of DDR1 in HUVECs infected with DDR1-EGFP adenovirus. Cell grown in gelatin-coated microfluidic chamber were subjected to laminar shear stress. (E) Quantification of DDR1 condensates number. Data were expressed as the means \pm SEM. $n=6$ biological replicates. Data were analyzed by Mann-Whitney test. Scale bars, 10 μ m.

• **This paper also needs a summary figure proposing how DDR1 is regulating mechanotransduction of shear.**

Response: We sincerely appreciate the reviewer's insightful recommendation concerning the inclusion of a summary figure illustrating how DDR1 is involved in mediating shear-induced mechanotransduction. In response to this suggestion, we have incorporated a graphical abstract that succinctly portrays the mechanistic interplay between DDR1 and shear mechanosensation and mechanotransduction. This summary figure has been integrated as Figure 8 in the manuscript.

Figure 8. DDR1 mechanosensation mediates the flow-activation of YAP and endothelial dysfunction.

The schematic depicts DDR1 as a primary mechanosensor in ECs, orchestrating cellular responses to shear flow. In areas of atheroprotective laminar flow, YAP phosphorylation prompts its binding with 14-3-3 proteins. This leads to YAP localization in the cytoplasm and subsequent degradation. Conversely, in atheroprone disturbed flow areas, DDR1 perceives the flow, initiating force-induced DDR1 oligomerization. This results in the formation of liquid-like biomolecular condensates involving DDR1 and 14-3-3. These condensates inhibit YAP phosphorylation and cytoplasmic sequestration, resulting in YAP activation and consequent endothelial dysfunction.

• **Lit review is not complete. Earlier studies on endothelial mechanotransduction should also be included in the introduction and discussion including the followings: DOI: 10.1111/brv.12814, DOI: 10.1111/brv.12814, DOI: 10.1111/brv.12814**

Response: We sincerely appreciate the reviewer's valuable suggestion to enhance the completeness of our literature review. In response to this recommendation, we have incorporated the provided references into both the introduction (Page 2, Line 11 in the manuscript) and discussion (Page 11, Line 3 in the manuscript) sections of the manuscript.

REVIEWERS' COMMENTS

Reviewer #1 (Remarks to the Author):

The manuscript has been improved with the addition of additional experiments and discussion.

Please add an explanation of why DDR1 in 1 does not inhibit DDR1 droplet aggregation to the manuscript text. This is potentially valuable information.

Reviewer #2 (Remarks to the Author):

I recommend acceptance of the paper in the current form. Authors have addressed all my comments, and I don't have any additional one.

Dear Reviewers,

Thank you very much for your time involved in reviewing the manuscript and your very encouraging comments on the merits.

Reply to Reviewer #1:

• Please add an explanation of why DDR1 in 1 does not inhibit DDR1 droplet aggregation to the manuscript text. This is potentially valuable information.

Response: Thanks for the valuable suggestion. It's noteworthy that despite treatment with OS and DDR1-IN-1, cells still displayed DDR1 droplets. This persistence can likely be attributed to the crucial role of the DDR1 transmembrane domain in driving DDR1 LLPS, as previously demonstrated in our research (*Circ Res.* 2023 Jan;132(1):87-105). The above explanation has been incorporated into the revised manuscript (Page 10, Lines 1~3).